# Pharmacological inhibition of PRMT7 links arginine monomethylation to the cellular stress response

Magdalena M. Szewczyk[1], Yoshinori Ishikawa[2,10], Shawna Organ[1,10], Nozomu Sakai[2,10], Fengling Li[1,10], Levon Halabelian [1], Suzanne Ackloo [1], Amber L. Couzens[3], Mohammad Eram[1], David Dilworth [1], Hideto Fukushi[2], Rachel Harding[1], Carlo C. dela Seña[1], Tsukasa Sugo[2], Kozo Hayashi[2], David McLeod [4], Carlos Zepeda[4], Ahmed Aman[4], Maria Sánchez-Osuna [5], Eric Bonneil [5], Shinji Takagi[2], Rima Al-Awar[4,6], Mike Tyers [5], Stephane Richard[7], Masayuki Takizawa[2], Anne-Claude Gingras [3], Cheryl H. Arrowsmith [1,8], Masoud Vedadi [1,6], Peter J. Brown [1], Hiroshi Nara[2✉] & Dalia Barsyte-Lovejoy [1,6,9✉]

Protein arginine methyltransferases (PRMTs) regulate diverse biological processes and are increasingly being recognized for their potential as drug targets. Here we report the discovery of a potent, selective, and cell-active chemical probe for PRMT7. SGC3027 is a cell permeable prodrug, which in cells is converted to SGC8158, a potent, SAM-competitive PRMT7 inhibitor. Inhibition or knockout of cellular PRMT7 results in drastically reduced levels of arginine monomethylated HSP70 family stress-associated proteins. Structural and bio-chemical analyses reveal that PRMT7-driven in vitro methylation of HSP70 at R469 requires an ATP-bound, open conformation of HSP70. In cells, SGC3027 inhibits methylation of both constitutive and inducible forms of HSP70, and leads to decreased tolerance for perturbations of proteostasis including heat shock and proteasome inhibitors. These results demonstrate a role for PRMT7 and arginine methylation in stress response.

[1] Structural Genomics Consortium, University of Toronto, Toronto, ON M5G 1L7, Canada. [2] Research, Takeda Pharmaceutical Company Limited, 26-1, Muraoka-Higashi 2-chome, Fujisawa, Kanagawa 251-8555, Japan. [3] Network Biology Collaborative Centre at the Lunenfeld-Tanenbaum Research Institute, 600 University Ave, Room 992, Toronto, ON M5G 1X5, Canada. [4] Drug Discovery Program, Ontario Institute for Cancer Research, Toronto, ON, Canada. [5] Institute for Research in Immunology and Cancer (IRIC) University of Montreal, 2950 Chemin de Polytechnique, Montreal, QC H3T 1J4, Canada. [6] Department of Pharmacology and Toxicology, University of Toronto, Toronto, ON M5S 1A8, Canada. [7] Terry Fox Molecular Oncology Group and Bloomfield Center for Research on Aging, Lady Davis Institute for Medical Research and Departments of Oncology and Medicine, McGill University, Montreal, QC H3T 1E2, Canada. [8] Princess Margaret Cancer Centre and Department of Medical Biophysics, University of Toronto, Toronto, ON M5G 2M9, Canada. [9] Nature Research Center, Vilnius, Akademijos 2, Lithuania. [10] These authors contributed equally: Yoshinori Ishikawa, Shawna Organ, Nozomu Sakai, Fengling Li. ✉email: nara@pharm.or.jp; d.barsyte@utoronto.ca

Protein arginine methyltransferases (PRMTs) methylate arginine residues in histone and non-histone proteins in a mono, and symmetric or asymmetric dimethyl manner[1,2]. Arginine methylation of both histone and non-histone substrates has major roles in transcription and chromatin regulation, cell signaling, DNA damage response, and RNA and protein metabolism[3,4]. PRMT7, a member of the PRMT family, has been functionally implicated in a wide range of cellular processes including DNA damage signaling[5–7], imprinting[8], and regulation of pluripotency[9–11]. Recently several elegant studies using *Prmt7* knockout mouse models also revealed the role of this methyltransferase in maintenance of muscle satellite cell quiescence, muscle oxidative metabolism, and B cell biology[12–14]. Although these studies have greatly expanded our understanding of PRMT7 biology, it remains an understudied member of the PRMT family with poor understanding of its cellular substrates.

PRMT enzymes display methylation preference for RGG/RG motifs enriched at protein–protein interfaces, whereas PRMT7 has been reported to target RXR motifs in arginine and lysine-rich regions[15,16]. PRMT7 is the sole evolutionary conserved class III PRMT enzyme, the subfamily which carries out only mono-methylation of arginine[17–19]. Other PRMT family members such as PRMT1 or PRMT5 catalyze arginine dimethylation in an asymmetric or symmetric manner, respectively, playing distinctly different downstream biological roles[1]. Remarkably, PRMT7-mediated monomethylation of histone H4R17 allosterically potentiates PRMT5 activity on H4R3[20]. Thus, possible overlap between substrates for PRMT7 and other PRMT enzymes and their interplay is complex and for most part still largely unknown. The best-characterized PRMT7 substrates are histone proteins, such as H3, H4, H2B, and H2A[1,3,6,18]. Additional non-histone PRMT7 substrates such as DVL3[21], G3BP2[22], and eukaryotic translation initiation factor 2 alpha (eIF2α)[23] have also been described. Proteomics studies have identified an abundance of cellular monomethyl arginine-containing proteins[24–27], however as other PRMT family members may be responsible for this methylation, it is not clear which of these substrates are dependent on PRMT7 as systematic studies of PRMT7 cellular substrates are lacking.

To enable further discovery of PRMT7 biology and to better explore its potential as a therapeutic target, here, we report a chemical probe of PRMT7 methyltransferase activity. SGC8158 is a potent, selective, and SAM-competitive inhibitor of PRMT7. To achieve cell permeability, we utilize a prodrug strategy where upon conversion of SGC3027 by cellular reductases, the active component, SGC8158, potently and specifically inhibits PRMT7-driven methylation of cellular substrates. A systematic screen of arginine monomethylated proteins dependent on PRMT7 in cells identifies several RG, RGG, and RXR motif proteins. HSP70 family members involved in stress response, apoptosis, and proteostasis are PRMT7 substrates in vitro and in cells. Our data shows that PRMT7 methylates HSPA8 (Hsc70) and HSPA1 (Hsp70) on R469, which resides in a highly conserved sequence in the substrate-binding domain. SGC3027 inhibits the PRMT7-driven methylation impacting the thermotolerance and proteostatic stress response in cells.

## Results

**PRMT7 chemical probe compound characterization**. *S*-adenosyl methionine (SAM) is a co-factor required by all methyltransferases. To identify potential PRMT7 inhibitors, a library of SAM-mimetic small molecules was assembled and screened by a scintillation proximity assay (SPA) against PRMT7 using $^3$H-SAM and a histone H2B peptide (residues 23–37) as a substrate (Supplementary Fig. 1a). This screening resulted in the identification of SGC0911 with $IC_{50}$ of 1 μM (Fig. 1a). Further derivatization of this hit yielded SGC8172, which displayed improved potency ($IC_{50} < 2.5$ nM) but was not selective for PRMT7 (Supplementary Table 1). Additional optimization of the methylene linker length between the terminal amine moiety of SGC0911 and the adenosine core structure resulted in SGC8158, a potent PRMT7 inhibitor ($IC_{50} < 2.5$ nM) (Fig. 1a, b), which showed good selectivity over a panel of 35 methyltransferases including PRMTs (Fig. 1c; Supplementary Tables 2 and 3) and kinases (Supplementary Fig. 2). Importantly, we also developed a negative control compound (SGC8158N) which was markedly less potent ($14 \pm 2$ μM) against PRMT7 (Fig. 1a, c) and other protein methyltransferases (Supplementary Table 2). Binding of SGC8158 to PRMT7 was also confirmed by surface plasmon resonance (SPR; Supplementary Fig. 1b, c). From kinetic fitting, a $K_D$ value of $6.4 \pm 1.2$ nM, $k_{on}$ of $4.4 \pm 1.1 \times 10^6$ M$^{-1}$ s$^{-1}$ and $k_{off}$ of $2.6 \pm 0.5 \times 10^{-2}$ s$^{-1}$ were calculated from triplicate experiments. We next investigated the mechanism of action (MOA) of SGC8158 by determining the $IC_{50}$ values at various concentrations of substrate (SAM and peptide). With increasing concentrations of SAM in the presence of a constant peptide concentration, we observed higher $IC_{50}$ values, which indicated a SAM-competitive pattern of inhibition (Supplementary Fig. 3). However, no change in $IC_{50}$ value was observed as the concentration of peptide was increased at fixed concentration of SAM indicating a noncompetitive pattern of inhibition with respect to peptide substrate (Supplementary Fig. 3).

Human PRMT7 shares 93% sequence identity with *Mus musculus* PRMT7 (*Mm*PRMT7), and the key residues involved in SAM and substrate-binding sites are conserved between the two species. To provide further insight into the mechanism of action of SGC8158, we solved the crystal structure of full-length *Mm*PRMT7 in complex with SGC8158 refined to 2.4 Å resolution (referred to here as *Mm*PRMT7_SGC8158). PRMT7 is a pseudo-dimer in nature, composed of catalytically active (N-terminal) and inactive (C-terminal) methyltransferase domains[28]. The structure showed that SGC8158, interacted only with the catalytically active N-terminal methyltransferase domain of PRMT7 in *Mm*PRMT7_SGC8158 (Fig. 1d). Clear electron density was observed for the ribosyl and biphenylmethylamine moieties of SGC8158 (Supplementary Fig. 4). The ribosyl moiety of SGC8158 is almost identical in position to that of SAH in SAH-bound *Mm*PRMT7 (*Mm*PRMT7_SAH), (PDB ID: 4C4A), explaining the SAM-competitive kinetics observed in our assays (Supplementary Fig. 2). The biphenylmethylamine moiety of SGC8158 extends toward the conserved THW loop region (residues Arg311 to Met315) and displaces the Trp314 side chain, to occupy a hydrophobic pocket composed of Trp282, Phe146, Tyr48, Met296, Arg44, and Arg311 side chains, and forms an edge-to-face π-stacking interaction with Trp282 (Fig. 1d). Compared to *Mm*PRMT7_SAH (PDB ID: 4C4A), the His313 and Trp314 residues in THW loop were distorted in SGC8158-bound *Mm*PRMT7 and could not be modeled (Fig. 1d, e). The PRMT-conserved THW loop is part of the active-site pocket and involved in substrate binding[29], but interestingly SGC8158 did not affect the peptide substrate binding (Supplementary Fig. 3). Comparison of the THW loop region in *Mm*PRMT7 with all other PRMTs showed that this loop has a variable length and conformation (Fig. 1f), suggesting that it likely has a role in the selectivity of SGC8158 for PRMT7. Taken together, these data indicate that SGC8158 is a potent and selective inhibitor in vitro that binds in the adenosine region of the SAM-binding pocket of PRMT7.

**Identification of PRMT7 substrates**. In order to evaluate pharmacological inhibition of PRMT7 in cells, we first sought to

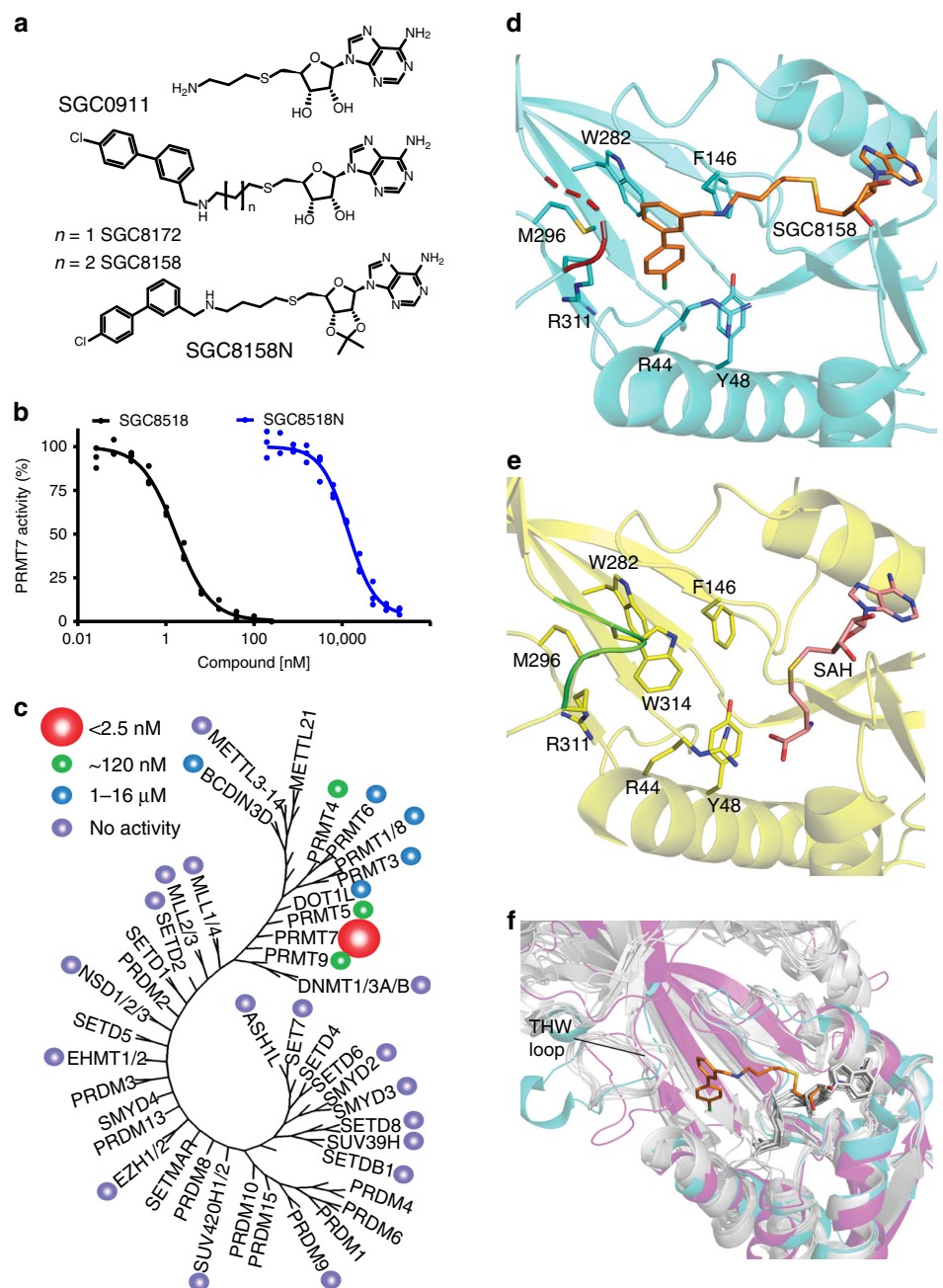

**Fig. 1 SGC8158 is a potent and selective PRMT7 inhibitor in vitro. a** Structures of HTS hit compound SGC0911, potent compound SGC8172, active component of the chemical probe SGC8158 and its negative control SGC8158N. **b** SGC8158 inhibits PRMT7 in vitro with IC$_{50}$ of <2.5 nM, whereas negative control compound SGC8158N has IC$_{50}$ of 14 ± 2 μM, ($n = 3$ biological replicates, mean ± SEM). **c** SGC8158 is selective against a panel of 35 proteins, DNA, and RNA methyltransferases. IC$_{50}$ values are represented by colored circles indicated top left of the panel. **d** Crystal structure of MmPRMT7 in complex with SGC8158. MmPRMT7 is shown in cartoon representation in cyan, hydrophobic pocket residues are shown in cyan sticks, and SGC8158 is in orange. The THW motif loop region is highlighted in red with dashed lines representing the unmodeled H313 and W314 residues. **e** For comparison, the crystal structure of MmPRMT7_SAH (PDB ID: 4C4A) is shown in cartoon representation in yellow, hydrophobic pocket residues are shown in yellow sticks, and SAH is in pink. The THW motif loop region is highlighted in green. **f** Comparison of the THW motif loop region of MmPRMT7_SGC8158 (in cyan) with that of PRMT5 (PDB ID: 5GQB) (in magenta), and PRMT1, 2, 3, 4, 6, and 8 (in gray) (PDB IDs: 1OR8, 5FUL, 2FYT, 2V74, 4Y30, and 5DST, respectively). SGC8158 is shown in orange sticks and SAH in gray sticks. Source data are provided as a Source Data file.

identify a cellular biomarker of its methylation activity. PRMT7 has a distinct substrate preference for RXR motifs surrounded by basic residues[17], and although RGG and RXR motifs are abundant in the proteome[16], very few have been validated in the cellular context. The fact that PRMT7 localization is mostly cytoplasmic (Fig. 2a, b), and most of the previously investigated substrates are histone proteins (i.e. nuclear), led us to undertake a

proteomics-based exploration of potential substrates of PRMT7. Wild-type (WT) and *PRMT7* knockout (KO) HCT116 cells were subjected to SILAC (stable isotope labeling by/with amino acids in cell culture) and monomethyl arginine immunoprecipitation followed by mass spectrometry analysis that included a targeted list of HSPA8 peptides (to ensure MS2 quantitation) within the data-dependent acquisition (DDA) cycle. Twenty-nine

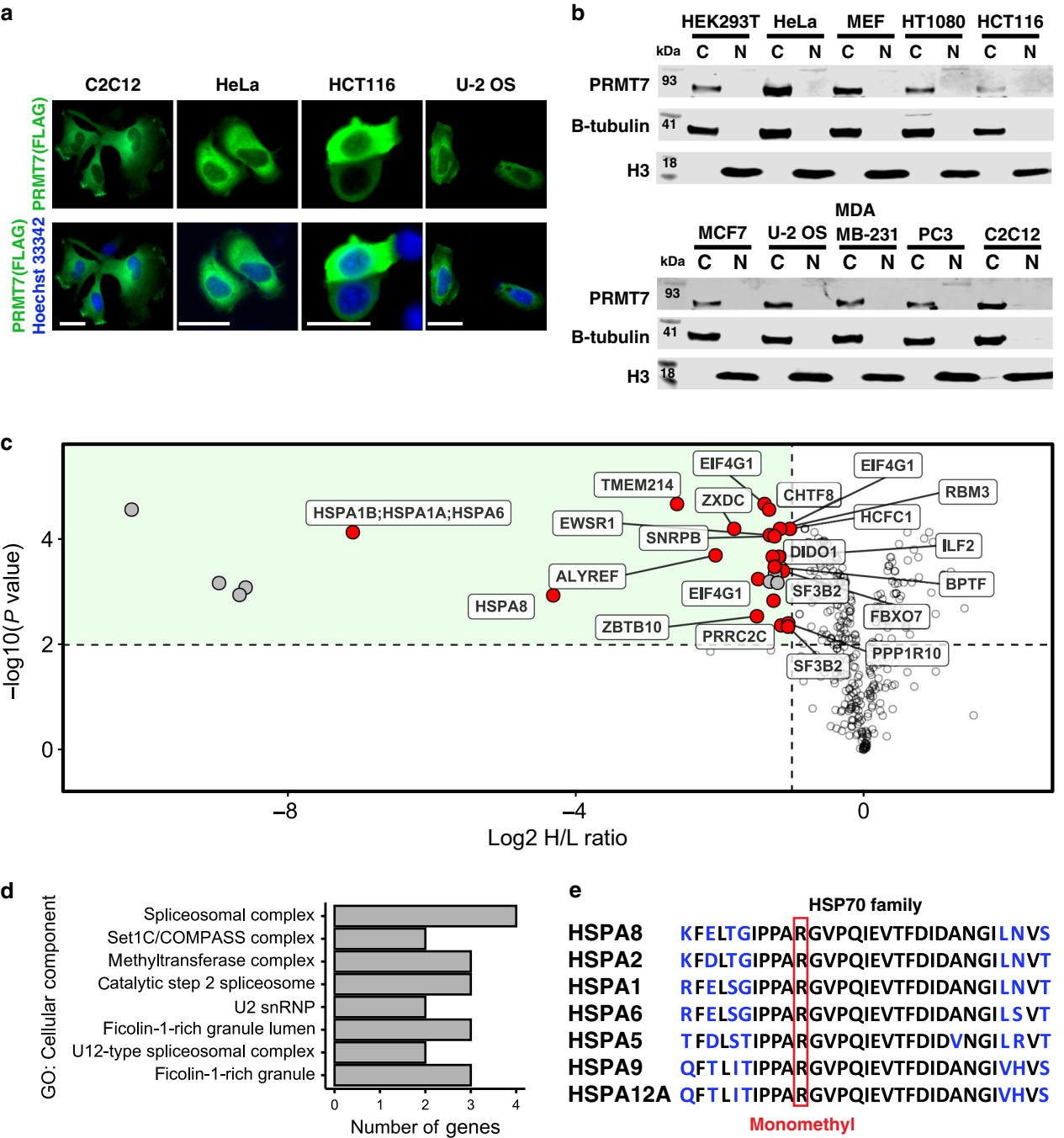

**Fig. 2 Identification of PRMT7 substrates. a** Localization of exogenous FLAG-tagged PRMT7 as analyzed by immunofluorescence. Scale bar—25 μm. Green—FLAG, Blue—Hoechst dye (nucleus). The experiment repeated independently three times with similar results. **b** Cellular fractionation of endogenous PRMT7 in several cell lines. C cytoplasmic fraction, N nuclear fraction. Tubulin indicates β-Tubulin control. The experiment was repeated independently twice for HEK293T cells with similar results. **c** Volcano plot showing log2 heavy/light ratio of SILAC-labeled monomethyl arginine peptides from WT (L, unlabeled) relative to *PRMT7* KO (H, heavy RK labeled) HCT116 cells. Dashed lines represent significance cut-offs of *H/L* ratio < −1 and adjusted *p*-value < 0.01 (*n* = 4). Labeled points, further highlighted in red, correspond to reported Rme1 sites found in the PhosphoSitePlus® v6.5.8[30]. *p*-values from four independent replicates calculated by empirical Bayes moderated *t*-tests and adjusted using the Benjamini–Hochberg procedure as implemented in the Bioconductor package limma (v3.38.3)[90]. **d** Cellular component gene ontology terms associated with 27 significantly depleted arginine methylation events in *PRMT7* KO relative to WT cells identified in **c**. **e** HSP family sequence alignment showing HSPA8 R469 resides in a highly conserved region where R469 is boxed in red. Source data are provided as a Source Data file.

significantly differentially methylated peptides representing 24 unique proteins were identified. Twenty-one peptides (from 18 proteins) were previously reported as arginine methylated[30] (highlighted in Fig. 2c, Supplementary Table 4). The analysis of total protein levels in *PRMT7* KO and WT cells showed no

significant change in protein abundance for the differentially methylated peptides indicating that the observed reduction in methylation was due to reduced monomethylation activity as opposed to perturbation of total protein levels (Supplementary Table 4). Most of the identified methylated proteins were

associated with RNA metabolism (Fig. 2d). For several proteins such as HSPA8, HSPA6/1A/B no detectable levels of R469 methylated peptides were found in the immunoprecipitated samples originating from the *PRMT7* KO cells, thus we performed validation and quantified their methylation in the input samples (Supplementary Fig. 5). This analysis showed that HSPA8 peptide FELTGIPPAPR-469 is highly methylated in a PRMT7-dependent manner in HCT116 cells. Sequences surrounding R469 are highly conserved among HSP70 family members (Fig. 2e), including the constitutively expressed HSPA8, and stress inducible HSPA6/HSPA1 proteins. The PRMT7-dependent HSP70 monomethylation was also detected using pan-monomethyl arginine antibody in HCT116 cell lysates (Supplementary Fig. 6a, b).

**PRMT7 methylates HSP70 proteins in cells**. In order to determine HSPA8 methylation in other cell systems, we investigated the endogenous regulation of HSP70 methylation in *PRMT7* KO (HCT116, C2C12 cells), siRNA knockdown (HEK293 and MCF7 cells) and mouse embryonic fibroblasts (MEFs) derived from WT or *Prmt7* KO mice. The *PRMT7* genetic knockout or knockdown resulted in decreased methylation associated with monomethyl arginine signal coinciding with the HSP70 specific signal (Fig. 3a, Supplementary Fig. 7). As HSP70 proteins are induced and have a role in heat shock response, we investigated whether the induced forms of HSP70, such as HSPA6, and HSPA1, are methylated by PRMT7. Heatshock exposure resulted in increased levels of the inducible HSP70 forms, coinciding with increased abundance of the arginine monomethylated signal indicating a matched rapid methylation by PRMT7 (Fig. 3b).

The re-expression of WT PRMT7 or the catalytic-dead mutant R44A in HCT116 *PRMT7* KO cells indicated that only WT PRMT7 was able to methylate HSP70 proteins (Fig. 3c). Moreover, introduction of WT or R469A mutant HSP70 into HCT116 *PRMT7* KO cells confirmed the methylation site of HSPA8/HSPA1 as identified in the proteomic analysis dataset (Fig. 3d). R469 has previously been identified as a methylation site for PRMT4 and PRMT7[31]; however, in MCF7 cells, we observed a major effect on HSP70 methylation driven by PRMT7 but not PRMT4 (Supplementary Fig. 7) possibly indicating cell-type-specific effects. These results demonstrate robust PRMT7-dependent methylation of both steady state and inducible HSP70 proteins in cells.

**PRMT7 methylates the open form of HSP70**. HSP70 proteins have dynamic structures with nucleotide and substrate-binding domains undergoing marked conformational changes upon nucleotide or substrate binding[32,33]. Posttranslational modification of various HSP70 proteins has been shown to modulate their activity and can alter the conformational landscape of modified chaperones[34–36]. As R469 resides in the substrate-binding domain, we examined the 3D protein structure to determine the accessibility of the R469 side chain. Available full-length structures of HSPA5 (65% sequence identity to HSPA8) and HSPA1A (86% identity) were analyzed for the positioning of R469. The structures of HSPA5 in both the ATP-bound open state[37] as well as the ADP-bound closed lid state[37,38] for the substrate binding and nucleotide binding domains, respectively, and HSPA1A[39] substrate-binding domain in the ADP-bound state were used. Analysis of these HSP70 structures indicated that R469 resides in a loop of the substrate-binding domain which is likely of limited accessibility in the ADP-bound form of HSP70 when the substrate-binding domain is in a closed conformation (Fig. 4a–c).

In order to establish whether PRMT7 methylates HSP70 in vitro, we performed full characterization of PRMT7 enzyme kinetics with HSPA8 as substrate. In agreement with structural analysis, our data indicated that PRMT7 methylates HSPA8 in the presence of ATP (Fig. 4d, e). Apparent kinetic parameters were then determined for HSPA8 methylation yielding a $K_m$ for SAM of $1.6 \pm 0.1\,\mu M$ (Fig. 4d), and $K_m$ for HSPA8 of $10.6 \pm 0.1\,\mu M$ (Fig. 4e). To test the specificity of this activity, we also tested the activity of PRMT7 on HSPA8 with a single arginine to lysine mutation (R469K). Consistent with previous findings, PRMT7 was completely inactive with the HSPA8–R469K mutant as substrate in the presence or absence of ATP (Fig. 4f). We further tested the effect of SGC8158 and SGC8158N on PRMT7-dependent methylation of HSPA8. SGC8158 inhibited PRMT7 methylation of full-length HSPA8 in vitro with $IC_{50} = 294 \pm 26$ nM under balanced conditions. As expected, SGC8158N showed very poor inhibition with an $IC_{50}$ value estimated to be higher than $100\,\mu M$ (Fig. 5a).

**SGC3027, a prodrug form of SGC8158, inhibits PRMT7 in cells**. Having established that PRMT7 methylates cellular HSP70, we returned to SGC8158 in order to determine its cellular activity. We observed no inhibition of cellular HSP70 methylation by this compound, likely due to its SAM-like structure and low cell permeability often associated with SAM analogs[40,41,42]. To increase the cellular permeability, we employed the Trimethyl Lock prodrug strategy in which SGC8158 is derivatized with a quinonebutanoic acid that masks a secondary amine group and increases lipophilicity[43]. The resulting derivative, SGC3027, undergoes reduction in cells followed by rapid lactonization, releasing the active component SGC8158 (Fig. 5b). The same strategy was employed for negative control prodrug compound SGC3027N (Supplementary Fig. 8c)

SGC3027 and SGC3027N prodrug compounds were efficiently converted into the active component, SGC8158 and SGC8158N, respectively, in cells (Supplementary Fig. 8). SGC3027 inhibited HSP70 methylation with $IC_{50}$ of $2.4 \pm 0.1\,\mu M$ (Fig. 5c, d) in C2C12 cells, and the inactive compound had no effect at $5\,\mu M$, the cellular $IC_{90}$ of SGC3027, and had a minimal effect at $10\,\mu M$ (Fig. 5e). In addition, SGC3027, but not SGC3027N, was effective at reducing HSP70 methylation in several commonly used cell lines (Supplementary Fig. 9). SGC3027 selectively inhibited PRMT7 but not PRMT1, 4, 5, 6, 9, and DOT1L (closest in vitro hits) at $5\,\mu M$ exposure (Supplementary Fig. 10). In *Prmt7* KO MEFs, SGC3027 does not affect the methylation of HSP70 (Supplementary Fig. 11), further confirming the on-target activity for PRMT7. Taken together these data demonstrate that SGC3027 is a selective and cell-active inhibitor of PRMT7.

**PRMT7-driven methylation is cytoprotective in proteostasis disruption**. HSP70 family members have key roles in nascent protein folding and refolding, as well as distinct roles in anti-apoptotic responses[44]. To determine whether PRMT7-driven methylation contributes to protection from cellular toxic insults, we investigated cell survival in response to thermal and proteasome stress using the genetic *Prmt7* KO models and a chemical biology approach employing the SGC3027 chemical probe. *Prmt7* KO MEF cells were more sensitive to acute heat stress with fewer cells surviving the treatment (Fig. 6a) and more cells undergoing apoptosis (Fig. 6b). SGC3027, but not the negative control SGC3027N, showed a similar response (Fig. 6c, Supplementary Fig. 12a), indicating that SGC3027 phenocopies the genetic ablation of *Prmt7*. We also investigated whether PRMT7 stress protection extends to proteasome stress using proteasome inhibitor bortezomib. Compared to WT cells, *Prmt7* KO MEF cells

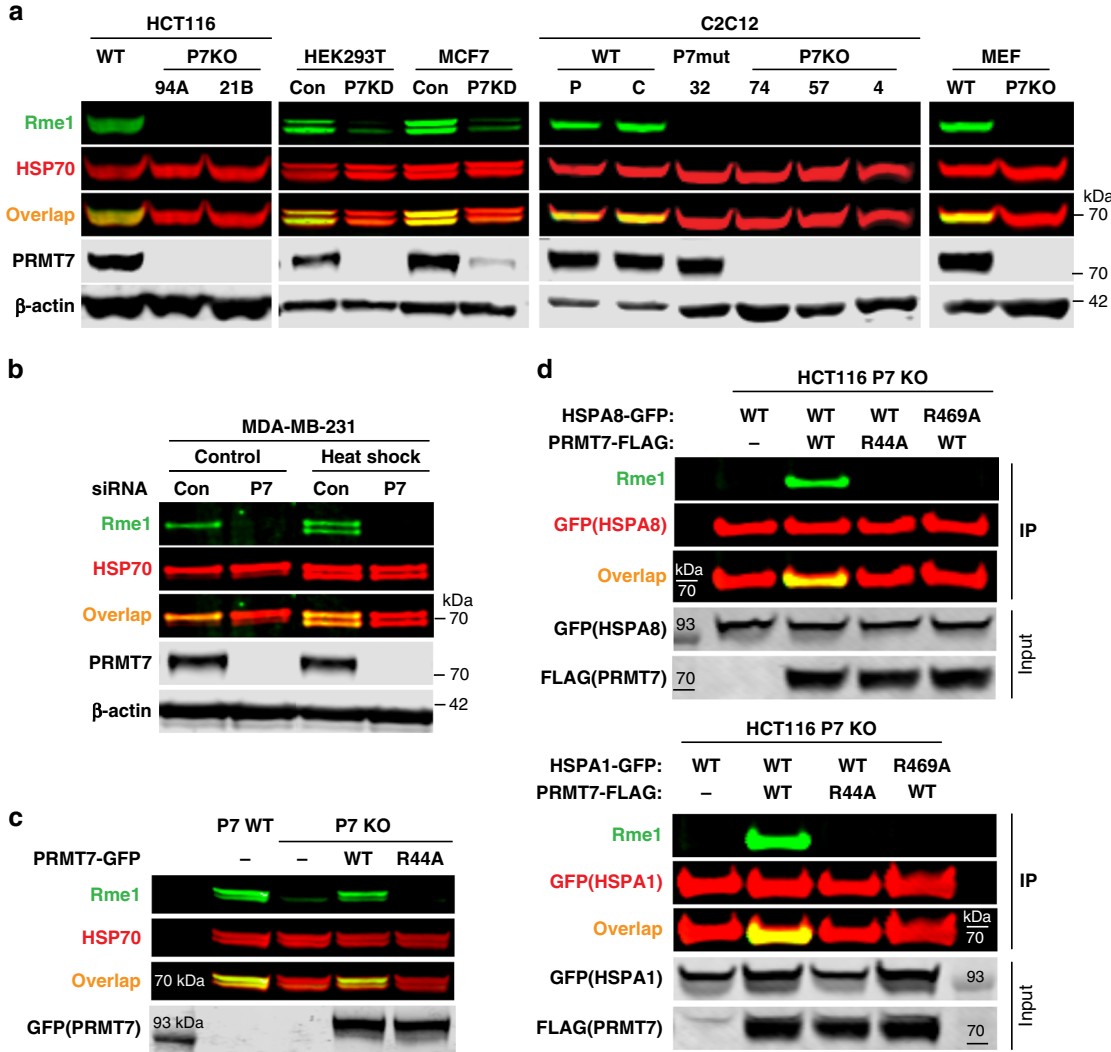

**Fig. 3 HSP70 R469 is methylated by PRMT7 in cells. a** *PRMT7* (P7) knockout (KO) or knockdown (KD) reduces HSP70 methylation in various cell lines. 94A, 21B—HCT116 CRISPR *PRMT7* KO clones; P parental C2C12; C-C2C12 expressing control guide RNA; 32 C2C12 CRISPR clone expressing PRMT7 catalytic mutant (delY35,A35S); 4,57,74—C2C12 CRISPR *Prmt7* KO clones. *PRMT7* was knocked down in HEK293T and MCF7 cells using siRNA, Con control, P7KD *PRMT7* knockdown. **b** Monomethylation of inducible and constitutive HSP70 is PRMT7-dependent. MDA-MB-231 cells were transfected with PRMT7 siRNA for 3 days, heat-shocked for 1 h at 42 °C and analyzed 24 h after heat shock. **c** Only wild-type PRMT7 is able to rescue the HSP70 arginine monomethylation in HCT116 *PRMT7* KO cells. Cells were transfected with GFP-tagged PRMT7 WT or catalytic mutant (R44A). **d** HSP70 R469A mutation blocks PRMT7-mediated methylation of HSPA8 and HSPA1. HCT116 *PRMT7* KO cells were co-transfected with FLAG-tagged *PRMT7* WT or R44A mutant and GFP-tagged HSPA8 or HSPA1 WT or R469A mutant. HSPA1/8-GFP was immunoprecipitated and analyzed for arginine monomethylation levels. The HSP70 methylation in MCF7, HCT116, and HEK293T cells was analyzed in cytoplasmic fraction to avoid unspecific band overlap. The experiments in **a**–**d** were repeated independently at least three times with similar results. Source data are provided as a Source Data file.

were more sensitive to acute bortezomib-induced cell death, having a poorer recovery after 4 or 20 h exposure to bortezomib (Fig. 6d, Supplementary Fig 12b). SGC3027, but not SGC3027N, sensitized the WT *Prmt7* MEFs to bortezomib, whereas neither compound affected *Prmt7* KO cells (Fig. 6e, f). These results indicate that PRMT7-driven methylation has a cytoprotective role in stress response, whereas inhibition of PRMT7 catalytic function can sensitize cells to toxic stimuli. To gain further insight into how R469 methylation may regulate HSP70 function, we investigated the effects of mutation (R469K) or PRMT7 inhibition on the functional properties of HSP70. R469K mutation did not affect HSP70 (HSPA8) driven ATPase activity (Supplementary Fig. 13) or the ability of HSPA8 to bind to the co-chaperones STIP1 (HOP) or STUB1 (CHIP) (Supplementary Fig. 14). The R469K mutation, however, reduced the ability of HSP70 to refold heat-denatured luciferase in cells (Supplementary Fig. 15) as well

as diminished the extent of stress granule prevention by HSP70 overexpression (Fig. 6g, h). SGC3027 consistently phenocopied the luciferase refolding and stress granule prevention effects of the mutation (Supplementary Fig. 15, Fig. 6g, h), further indicating selective modulation of cellular PRMT7 function. Thus, PRMT7 methylation of HSP70 proteins impacts the function of HSP70 in the cellular stress response.

## Discussion
PRMT7 is a monomethyl arginine methyltransferase that has a role in muscle physiology and stem cell biology[3,10–13]. However, the PRMT7 substrates that mediate this biology are not well understood, and tools to selectively and temporally modulate PRMT7 catalytic activity are lacking. Here we report the selective PRMT7 chemical probe, SGC3027, and a closely related but inactive compound, SGC3027N, for use as a specificity control for

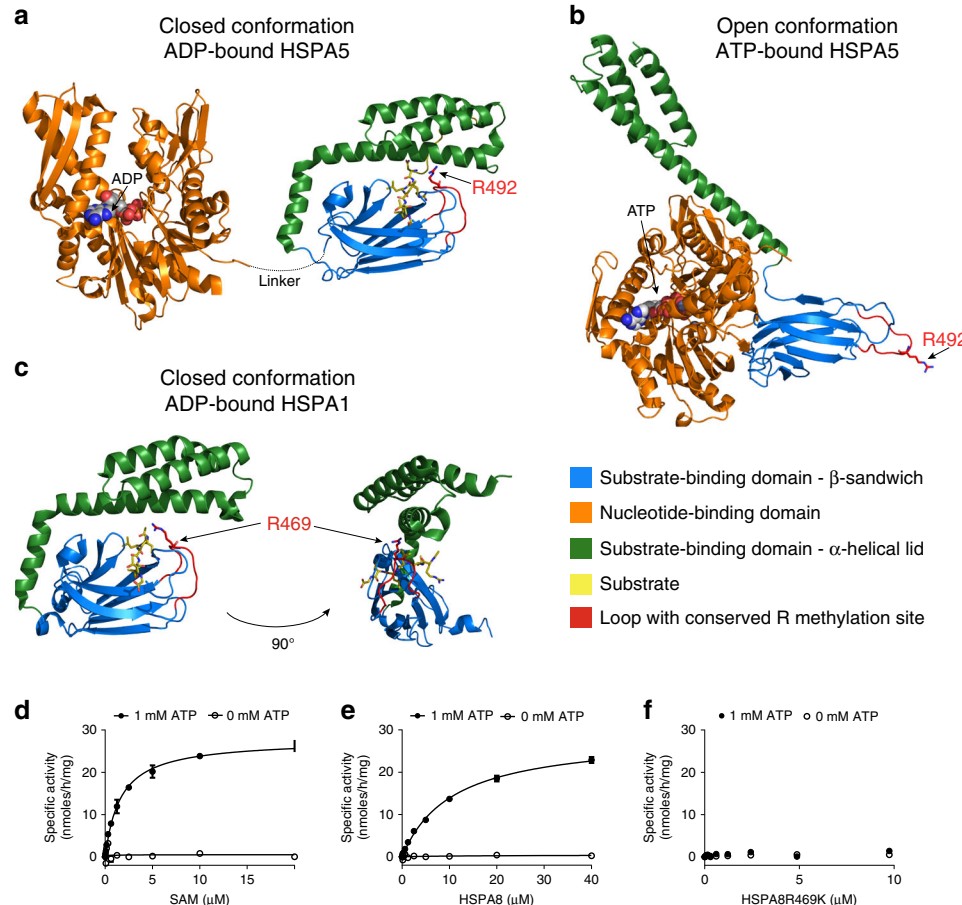

**Fig. 4 PRMT7 monomethylation of HSP70 depends on the open (ATP-bound) form of HSP70. a–c** HSP70 structures in closed and open confirmations reveal differential accessibility of the conserved R469-containing sequence (HSPA8) monomethylated by PRMT7. Structures are color coded for domains (orange—ATP binding, blue—substrate binding, and green—lid domains). The HSP70 substrate-binding domain loop containing PRMT7 methylated arginine is colored red. **a**, **b** Closely related homolog HSPA5 structures (65% overall sequence identity to HSPA8) were analyzed to investigate the position of the arginine methylation site in the different conformations. In the ADP-bound state, the lid of the substrate-binding domain is closed (PDB 5E85), limiting accessibility of the R492 (analogous to R469 in HSPA8) residue for methylation by PRMT7. In the ATP-bound form (PDB 5E84), the arginine residue is accessible therefore permitting access by the PRMT7 enzyme. **c** The structure of the more closely HSPA8 related HSPA1A (86% overall sequence identity and 82% sequence identity for aa. 386–646 in the substrate-binding domain, PDB 4PO2) in the closed conformation in which R469 is occluded by the lid subdomain. **d–f** Kinetic analysis of HSPA8 methylation by PRMT7 in vitro. Kinetic parameters were determined for HSPA8 methylation in the presence and absence of ATP. PRMT7 had no activity in the absence of ATP. **d** Kinetic analysis at fixed 10 μM HSPA8 (SAM $K_m = 1.6 \pm 0.1$ μM). **e** Kinetic analysis at fixed 20 μM of SAM (HSPA8 $K_m = 10.6 \pm 0.1$ μM and $k_{cat}$ of $2.2 \pm 0.1$ h$^{-1}$). **f** HSPA8–R469K mutant is not methylated by PRMT7 in vitro. The results are mean ± SEM of three technical replicates. Source data are provided as a Source Data file.

biological experiments. We also identified PRMT7 substrates using proteomics and further validated HSPA8 and related HSP70 family members as PRMT7 substrates by employing in vitro assays, genetic methods, and chemical biology. Our data suggest that PRMT7 activity has a role in HSP70 protein function, cellular thermotolerance, and proteasomal stress response. Therefore, SGC3027 may be a useful modulator of cellular proteostasis in stress response under physiological and pathological conditions.

SGC8158 is a structural derivative of SAM, acts as a SAM-competitive inhibitor of PRMT7, and occupies the adenosine pocket in the SAM-binding site of PRMT7. Other potent and selective SAM-like inhibitors of methyltransferases including EPZ004777[45] and SGC0946 for DOT1L[46], and LLY-283 for PRMT5[47] feature extensively modified adenosine and methionine moieties that likely enhance the cell permeability of otherwise cell impermeable SAM. We employed an alternative prodrug strategy in which adding the quinonebutanoic moiety increased cell permeability and allowed for cellular reductases to generate the

active compound, SGC8158. Notably, the cellular reductase-driven activation of SGC3027 to the active SGC8158 may vary among cell types depending on the abundance or activity of reductase enzymes. Thus, we recommend that the appropriate concentration range for use of these probes should be empirically evaluated in each experimental setting by monitoring a biomarker such as HSP70 methylation before evaluation of specific biological readouts.

SGC3027 sensitized cells to heat and proteasomal stresses indicating that PRMT7 catalytic function is required for normal physiological response to these stimuli. *Prmt7* KO cells were also more sensitive to these stresses, indicating that the chemical probe phenocopies the genetic knockout effects. Heat stress and proteasome inhibition elicits orchestrated protective responses, with the central goal of maintenance of cell protein homeostasis, proteostasis. Failure to maintain proteostasis results in numerous diseases. For example, altered proteostatic balance due to rewiring of chaperome complexes has been observed in cancer cells, contributing to drug resistance[48,49].

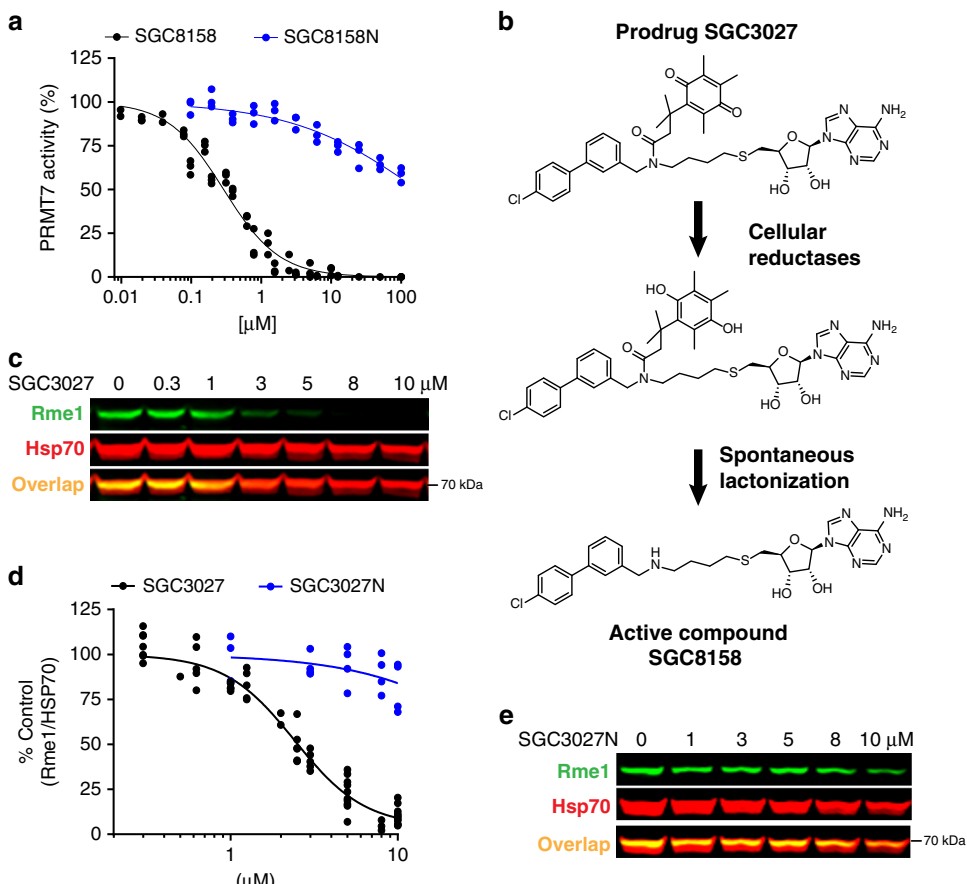

**Fig. 5 SGC3027 inhibits HSP70 methylation in cells and its active component SGC8158 methylation in vitro. a** SGC8158 inhibits PRMT7 methylation of HSPA8 in vitro. SGC8158 $IC_{50} = 294 \pm 26$ nM (2 biological replicates, each with technical $n = 3$, mean $\pm$ SEM), SGC8158N $IC_{50} > 100$ μM ($n =$ three technical replicates). The methylation assay was performed in the presence of ATP ($n = 3$ technical replicates). **b** SGC3027 is a prodrug cellular inhibitor of PRMT7 as illustrated by the prodrug conversion to the active component in cells. **c** SGC3027 inhibits PRMT7-dependent HSP70 monomethylation in C2C12 cells. Cells were treated with the compound for 2 days. The experiment was repeated four times with similar results. **d** Quantification of SGC3027 and SGC3027N effects on HSP70 monomethylation in C2C12 cells. The graphs represent non-linear fits of Rme1 signal intensities normalized to intensities of HSP70. SGC3027: $n = 11$, four separate experiments, $IC_{50} = 2.4 \pm 0.1$ μM; SGC3027N: $n = 4$ technical replicates, $IC_{50} > 40$ μM (mean $\pm$ SEM). **e** A representative blot for SGC3027N effects on HSP70 methylation. Rme1—arginine monomethylation. The experiment with 3 and 10 μM SGC3027N concentration was repeated three times with similar results. Source data are provided as a Source Data file.

PRMT7-dependent protection against cellular stress may have physiological importance in cancer cell survival, consistent with higher levels of PRMT7 that have been reported in breast cancer cell lines[50,51]. Interestingly, *PRMT7* was identified in a screen for sensitization to topoisomerase inhibitors in cancer cells[5], suggesting a wider range of stressors against which PRMT7 may be protective. Our findings of PRMT7 inhibition leading to the sensitization of cells to bortezomib-induced cell death indicate potential therapeutic applications in cancers such as multiple myeloma and some lymphomas. It is possible that PRMT7 inhibition may lead to overcoming resistance to bortezomib or other therapies. Further work is needed to determine whether PRMT7 inhibition can indeed synergize with proteasome inhibitor drugs in therapeutically relevant cell systems, and these efforts will be aided by the use of SGC3027 as a chemical probe for PRMT7.

The majority of PRMT7 substrates identified in this study were functionally implicated in RNA metabolism and splicing. Previous arginine methylome studies have also noted the abundance of mono and dimethyl arginine posttranslational modifications (PTMs) in proteins involved in splicing, transcription, and RNA metabolism[24–27,52,53]. Although about half of the PRMT7-dependent methylarginine peptides have been described in

various studies[24–27,54], several identified arginine methylation sites have not been previously reported, for example, the putative acetyltransferase NAT16, however, the related NAT10 and NAT6 are monomethylated[54]. Highlighting the complex relationship between PRMT enzyme activities, the PRMT7 methylation sites identified in several proteins, EIF4G1, ALYREF, RBM3, SF3B2, HSPA8, EWSR1, and SNRPB, were also reported as PRMT1, PRMT4, and PRMT5 sites[30,31,55–57]. We have chosen to focus on high confidence methylation of the HSP70 family as cellular substrates of PRMT7 leading us to further investigate the role of PRMT7 in stress response.

HSP70 R469 monomethylation by PRMT4 and PRMT7 was recently reported as contributing to transcriptional activation by retinoic acid receptor RAR[31]. Interestingly, PRMT4 methylation of HSP70 did not seem to require an ATP-driven conformational change[31], while we found that PRMT7-driven methylation occurs when HSP70 adopts an open, ATP-bound conformation that promotes R469 accessibility. In all of the examined structures of the substrate and ADP-bound forms of HSP70, R469 packs into the lid subdomain of the substrate-binding domain to ensure flexible, yet stable, interaction with the client protein[39], but rendering this residue inaccessible to modifying enzymes such as PRMT7. Although the substrate-binding domains and lid

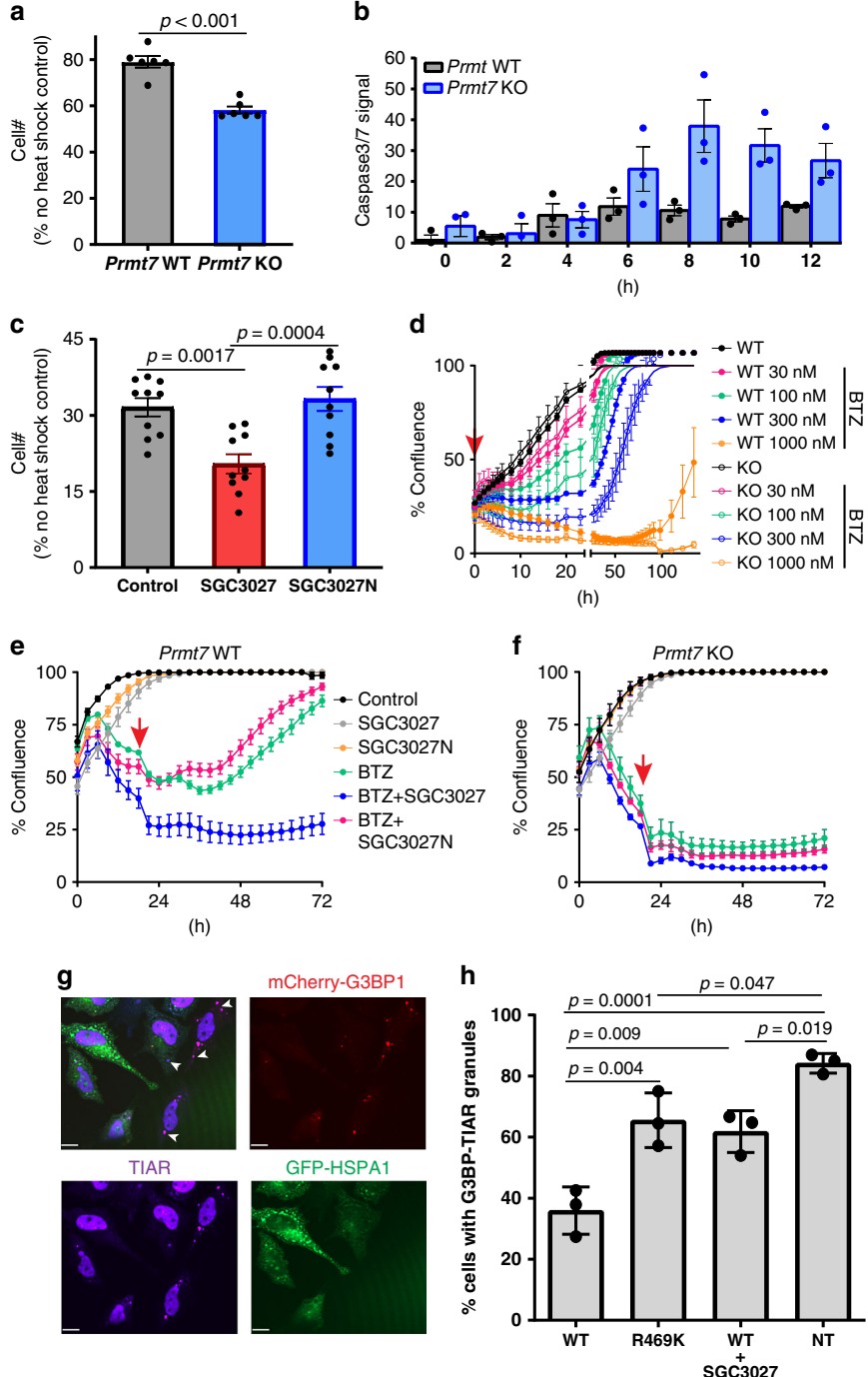

**Fig. 6 PRMT7 knockout/inhibition affects cell survival after heat shock or proteasomal stress. a**, **b** Loss of PRMT7 decreases cell survival and increases apoptosis levels after heat shock. MEF cells were heat-shocked for 20 min at 44 °C. Apoptosis was monitored immediately after the heat shock and cell number was determined 24 h later. The results shown are mean ± SEM of two biological replicates, each technical triplicate (**a**) and three technical replicates (**b**). Statistical significance was determined with unpaired Student t-test (two-tailed). **c** SGC3027 inhibition of PRMT7 activity decreases cell survival and increases apoptosis levels after heat shock. Cells were pretreated with 3 μM compounds for 2 days before heat shock. Cell number was determined as in **a**. The results shown are mean ± SEM of two biological replicates, each technical n = 5. Statistical significance was determined with one-way ANOVA with Tukey's post-hoc test. **d** Loss of PRMT7 decreases cell survival after bortezomib (BTZ) treatment. BTZ was removed after 4 h and the cell confluence was monitored 4 h after BTZ treatment. The results are mean ± SEM of 3–6 technical replicates. **e**, **f** SGC3027 decreases cell survival after BTZ treatment (30 nM) in *Prmt7* WT MEF (**e**) but not in *Prmt7* KO MEF (**f**). Cells were pretreated with 4 μM compounds for 2 days before BTZ treatment. After 20 h, BTZ was removed and SGC3027 or SGC3027N were replaced. The results are mean ± SD of 6–8 technical replicates. Red arrow indicates the time BTZ was removed. Confluency is a measure of cell number. **g** Overexpression of WT HSPA1 (HSP70/HSP72) inhibits the induction of G3BP-mcherry and TIAR-positive stress granules in response to proteasome inhibition. Scale bar is 14 μm, arrow indicates stress granules. **h** Quantitation of G3BP-mcherry and TIAR stress granules in GFP-positive cells overexpressing WT GFP-HSPA1 (with/without 3 μM SGC3027) or the catalytic mutant GFP-HSPA1 R469K and non-transfected cells (NT). The results are mean ± SD of three biological replicates. Statistical significance was determined with one-way ANOVA with Tukey's post-hoc test. Source data are provided as a Source Data file.

subdomains are more variable among HSP70 proteins than other regions, possibly due to the need of accommodating a large number of substrates[39], the region surrounding R469 is highly conserved among HSP70 family members and species. PTMs of the HSP70 lid and tail domains have been reported to disrupt the interaction with co-chaperone CHIP TPR[58]. Although the R469-containing loop resides in close proximity to the CHIP binding region, the R469K mutation did not affect the binding of CHIP.

The cytoprotective function of HSP70 proteins has been attributed to the modulation of protein refolding and transport of client proteins, as well as direct regulation of apoptotic signaling pathways[48,59]. Several HSP70 inhibitors have been reported and their utility is being explored in counteracting bortezomib resistance in multiple myeloma as well as therapeutic applications in other cancers[60–63]. PTMs of the HSP70 family members such as, lysine methylation, ubiquitination, acetylation, and phosphorylation have been reported[64], and some of the PTMs stabilize the HSP70/HSP90/HSP40 and client protein complexes allowing the formation of antiparallel HSP70 dimers[65]. K561 trimethylation by METTL21A and dimethylation by SETD1A have been reported to result in the modulation of HSP70 affinity for client proteins or potentiation of AURKB activity, respectively[66–68]. Here, using genetic and pharmacological means, we identify R469 mono-methylation as an abundant cellular modification that depends on PRMT7 catalytic function and correlates with the cytoprotective properties of PRMT7. Recently, it was reported that PRMT7 interacts and methylates eukaryotic translation factor eIF2α and regulates stress granule formation in response to various stresses[23]. Interestingly, EIF4G1 that is methylated by PRMT7 and PRMT1[56] also has a role in translation and stress granules[69–71]. Stresses such as proteasome inhibition induce the stress granules in an eIF2α phosphorylation-dependent manner[72], whereas HSP70 proteins that modulate and prevent the stress granule formation have important roles in their dynamic regulation[72,73]. Our data indicate that HSP70 R469 methylation by PRMT7 is important for stress granule response upon proteasome inhibition and provides additional links to the complex interplay of proteasome, translation, and chaperone systems acting on stress granules to ensure cell survival.

We uncovered aspects of PRMT7 biology associated with proteostasis, identified PRMT7 cellular substrates and validated HSP70 family members HSPA1/6/8 as PRMT7 substrates, whose methylation is likely to contribute to the cytoprotective and stress response function of PRMT7. SGC3027, together with its negative control SGC3027N, will be useful tools for further understanding PRMT7 function in physiological and disease states.

## Methods

**Protein expression and purification.** Full-length PRMT7 was expressed in Sf9 cells grown in HyQ® SFX Insect serum-free medium (ThermoScientific). Cells were harvested and lysed, and the cleared lysate was incubated with 5 mL anti-FLAG M2-Agarose (Sigma) in 50 mM Tris–HCl, pH 7.5, 150 mM NaCl, 10% glycerol and washed with the same buffer with 500 mM NaCl. The pure recombinant protein was eluted from the column using the same buffer with 0.1 mg/mL FLAG peptide (Sigma). Pure PRMT7 was flash frozen and stored at −80 °C. Full-length HSPA8 was overexpressed in *Escherichia coli* strain BL21(DE3) V2R-pRARE2 during an overnight induction with 0.5 mM isopropyl 1-thio-D-galactopyranoside at 18 °C. Cells were suspended in 20 mM Tris–HCl (pH 8.0, 1 mM DTT, 300 mM NaCl). The clarified lysate was loaded onto a Hispur™ nickel-nitrilotriacetic acid column (ThermoScientific) and washed with buffer. Then, protein was eluted and concentrated. Protein purity was determined by SDS-PAGE and liquid chromatography–mass spectrometry (LC–MS).

**Radioactive activity assay in vitro.** Assays using biotinylated H2B (23–37) as a substrate were performed in buffer (20 mM Tris–HCl, pH 8.5, 0.01% Tween-20, and 5 mM DTT) containing 5 nM PRMT7, 1.1 μM ³H-SAM (Cat.# NET155-V250UC; Perkin Elmer; www.perkinelmer.com) and 0.3 μM H2B (23–37). The reaction mixtures were incubated for 60 min at 23 °C. To stop the enzymatic reactions, 10 μL of 7.5 M guanidine hydrochloride was added, followed by 180 μL

of buffer (20 mM Tris, pH 8.0), mixed and then transferred to a 96-well FlashPlate (Cat.# SMP103; Perkin Elmer; www.perkinelmer.com). After mixing, the reaction mixtures in Flash plates were incubated for 1 h and the CPM were measured using Topcount plate reader (Perkin Elmer, www.perkinelmer.com). The CPM counts in the absence of compound for each dataset were defined as 100% activity. In the absence of the enzyme, the CPM counts in each dataset were defined as background (0%). The IC50 values were determined using GraphPad Prism 7 software. For the kinetic analysis of HSPA8 methylation by PRMT7, the assay mixture contained 20 mM Tris–HCl, pH 8.5, 0.01% Tween-20, and 5 mM dithiothreitol (DTT), 1 mM MgCl₂, 1 mM ATP where indicated, 250 nM PRMT7, fixed concentration (20 μM) of SAM, various concentrations (up to 40 μM) of HSPA8; or fixed concentration (10 μM) of HSPA8 with different concentrations of SAM (up to 20 μM). IC50 determinations of SGC8158 and SGC8158N were performed at 150 nM PRMT7, close to $K_m$ values of both SAM (2 μM) and substrate HSPA8 (7 μM). To determine the mode of action, the experiments were performed in the presence of fixed biotinylated H2B peptide (residues 23–37) or HSPA8 substrate concentration and increasing SAM concentration or at fixed concentration of SAM and varying concentration of the substrate. Twenty μL of reaction mixtures were incubated at 23 °C for 60 min. To stop reactions, 100 μL of 10% trichloroacetic acid (TCA) was added, mixed and transferred to filter-plates (Millipore; cat.# MSFBN6B10; www.millipore.com). Plates were centrifuged at 930 × *g* (Allegra X-15R—Beckman Coulter, Inc.) for 2 min followed by two additional 10% TCA washes and one ethanol wash followed by centrifugation. Plates were dried and 30 μL MicroScint-O (Perkin Elmer) was added to each well, centrifuged and removed. 50 μL of MicroScint-O was added again and CPM was measured using Topcount plate reader. The IC50 values were determined using GraphPad Prism 7 software.

**SPR analysis.** SPR analysis was performed using a Biacore™ T200 (GE Health Sciences Inc.) at 20 °C. Approximately 5500 response units of Bio-PRMT7 (amino acids 1–692) was fixed on a flow cell of a SA chip according to manufacturer's protocol, whereas another flow cell was left empty for reference subtraction. SPR analysis was performed in HBS-EP (20 mM HEPES pH 7.4, 150 mM NaCl, 3 mM EDTA, 0.05% Tween-20) with 3% DMSO. Five concentrations of SGC8158 (150, 50, 16.6, 5.5, and 1.85 nM) were prepared by serial dilution. Kinetic analysis was performed using single cycle kinetics with contact time of 60 s, off time of 300 s, and a flow rate of 100 μL min⁻¹. To favor complete dissociation of compound for the next cycle, a regeneration step (300 s, 40 μL min⁻¹ of buffer), a stabilization period (120 s) and two blank cycles were run between each cycle. Kinetic curves were fitted using a 1:1 binding model and the Biacore™ T200 Evaluation software ver 3.1 (GE Health Sciences Inc.).

**Selectivity assays.** The methyltransferase selectivity was assessed at two compound concentrations of 1 and 10 μM by radiometric assays using tritiated-SAM. For methyltransferases: MLL1, MLL3, EZH1 (PRC2), and EZH2 (PRC2), G9a, GLP, SUV39H1, SUV39H2, SUV420H1, SUV420H2, SETD2, SETD8, SETDB1, SETD7, PRMT1, PRMT3, PRMT4, PRMT5/ MEP50 complex, PRMT6, PRMT7, PRMT8, PRMT9, PRDM9, SMYD2, SMYD3, DNMT1, and BCDIN3D the incorporation of a tritium-labeled methyl group into biotinylated substrate was monitored using scintillation proximity assay (SPA). Briefly, a 10 μL reaction containing ³H-SAM and substrate at concentrations close to the apparent $K_m$ values for each enzyme (balanced conditions) was prepared. The reactions were quenched with 10 μL of 7.5 M guanidine hydrochloride; 180 μL of 20 mM Tris buffer (pH 8.0) were added, and the mixture was transferred to a 96-well FlashPlate and incubated for 1 h. The counts per minute (CPM) was measured on a TopCount plate reader. The CPM in the absence of compound or enzyme was defined as 100% activity and background (0%), respectively, for each dataset[74–76]. Selectivity of SGC8158 against 342 kinases was evaluated in a kinase panel TR-FRET assay at 1 and 0.1 μM compound in duplicates as described before[77]. Briefly, tagged kinases were incubated with compound and staurosporine-based fluorescent probe where the binding was detected using Tb conjugated anti-tag antibody energy transfer to the probe. Excitation wavelength was 337 nm and fluorescent emission signal was measured for Tb (486 nm) and fluorescent probe (BODIPY, 515 nm). Specific and non-specific binding was determined in the absence and presence of unlabeled staurosporine.

**Recombinant MmPRMT7 expression and structure determination.** The full-length *M. musculus Prmt7* gene was cloned into a pFBOH-MHL baculovirus expression vector encoding an N-terminal His6 tag followed by tobacco etch virus protease (TEV) cleavage site and expressed in Sf9 cells. The recombinant mPRMT7 protein was first affinity purified with TALON beads followed by size-exclusion chromatography using S200 column pre-equilibrated with 20 mM Tris–HCl [pH 8.0] and 150 mM NaCl. The peak corresponding to the monomeric mPRMT7 was then incubated overnight with TEV protease. The cleaved fraction was then further purified to homogeneity by ion-exchange chromatography using Source Q column equilibrated with buffer A: 20 mM Tris–HCl [pH 7.5] and eluted with linear salt gradient of buffer B: 20 mM Tris–HCl [pH 7.5] and 1 M NaCl. As co-crystallization trials for MmPRMT7 with SGC8158 and Apo-MmPRMT7 failed to yield any crystals, we first generated *S*-adenosyl-L-homocysteine (SAH)-bound MmPRMT7

co-crystals by mixing MmPRMT7 (at 8 mg mL$^{-1}$) with 3-fold molar excess of SAH and setting vapor-diffusion sitting drops in a precipitant solution containing 2% (v/v) Tacsimate pH 6.0, 0.1 M Bis-Tris pH 6.5, 20% (w/v) PEG 3350. The SAH-bound MmPRMT7 crystals were then soaked into a 1 μL reservoir drop supplemented with 1 mM SGC8158 (dissolved from a previously prepared 100 mM DMSO stock solution), and 5% (v/v) glycerol for 3 days at room temperature. Crystals were then cryoprotected by displacing the precipitant solution with a paratone and cryo-cooled in liquid nitrogen. The MmPRMT7_ SGC8158 dataset was collected at the 24-ID-E beamline at the Advanced Photon Source (APS). Dataset was processed with HKL3000[78]. Initial phases were obtained by using MmPRMT7 (PDB ID:4C4A) as initial model in Fourier transform with refmac5 (version 5 5.8.0238)[79]. Model building was performed in COOT (version 0.8.9.2)[80] and the structure was validated with Molprobity (version 5 5.8.0238)[81]. SGC8158 restraints were generated using Grade Web Server (http://grade.globalphasing.org). Images were prepared with PyMol Software (Molecular Graphics System, v2.2.0, Schrödinger, LLC). (www.pymol.org). Crystallographic data collection and refinement statistics are provided in Supplementary Table 6.

**Constructs, cells, and antibodies**. MCF7 (ATCC® HTB-22™), C2C12, MEF WT and MEF *Prmt7* KO (kind gift from Dr. Stephane Richard, McGill University), U-2 Os (ATCC®HTB-96™), HT-1080 (ATCC®CCL-121™) and HEK293T (kind gift from Sam Benchimol, York University, ATCC®CRL-3216™), HeLa (ATCC®CRM-CCL-2™) were grown in DMEM (Wisent), HCT116 WT (ATCC®CCL-247™), in McCoy's (Gibco) and THP-1 (kind gift from Dr. Mark Minden, Princess Margaret Cancer Center, ATCC® TIB-202™), MDA-MB-231 (ATCC® HTB-26™) in RPMI1640 (Wisent) supplemented with 10% FBS (Wisent), penicillin (100 U mL$^{-1}$) and streptomycin (100 μg mL$^{-1}$). All mammalian cell lines were purchased from Cedarlane and Sf9 cells (#11496015) from ThermoFisher Scientific. Anti-Rme1 (#8015, 1:1000), anti-Rme2s (#13222, 1:2000), anti-mouse IgG Alexa Fluor 488 (#4408, 1:1000), anti-TIAR (#8509, 1:2000), and anti-rabbit IgG Alexa647 (#4414, 1:2000) were purchased from Cell Signaling Technologies. Anti-Hsp/Hsc70 was from Enzo (#ADI-SPA-820, 1:2000). Antibodies for PRMT7 (#ab179822, 1:1000), PRMT5 (#ab109451, 1:5000), H3K79m2 (#ab3594, 1:2000), H4 (#ab174628,1:2000), and β-actin (#ab3280,1:3000) were purchased from Abcam. Anti-PRMT4 (#A300-421A, 1:2000) was from Bethyl. Anti-GFP (#632381, 1:3000) used for western blot was purchased from Clontech. Anti-GFP used for IP was purchased from Invitrogen (#G10362, 1:200). Anti-Flag (#F4799, 1:5000) was from Sigma. Anti-SmBB' (#sc-130670, 1:100) and anti-BAF155 (#sc-32763, 1:200) was from Santa Cruz Biotechnologies. Anti-BAF155-R1064me2a (#ABE1339, 1:3000) was from Millipore. Anti-H4R3me2a (#39705, 1:2000) was from Active Motif. Goat-anti-rabbit IgG-IR800 (#926-32211, 1:5000) and donkey anti-mouse IgG-IR680 (#926-68072, 1:5000) were purchased from LiCor. Antibody recognizing methylated SAP145 was kind gift from Dr. Yanzhong Yang, Beckman Research Institute (1:1000). Full length of HSPA8, HSPA1 were cloned into pAcGFPN3 vector (Clontech) and PRMT7 were cloned into pAcGFPN3 (Clontech) or pcDNA3 (N terminus FLAG). Site-directed mutagenesis to generate PRMT7 R44A mutant, HSPA1 R469A and HSPA8 R469A mutants was performed using Q5®Site-Directed Mutagenesis Kit (NEB), following manufacturer's instructions. MEF WT and MEF *Prmt7* KO were immortalized at passage 3 by transfection of SV40LT.

**PRMT7 cellular assay**. C2C12 cells were plated and next day treated with compounds. After 48 h, cells were lysed in lysis buffer (20 mM Tris–HCl pH 8, 150 mM NaCl, 1 mM EDTA, 10 mM MgCl$_2$, 0.5% Triton X-100, 12.5 U mL$^{-1}$ benzonase (Sigma), complete EDTA-free protease inhibitor cocktail (Roche)). After 2 min incubation at RT, SDS was added to final 1% concentration. Cell lysates were analyzed in western blot for unmethylated and monomethylated Hsp70/Hsc70 levels. The IC$_{50}$ values were determined using GraphPad Prism 7 software.

**PRMT1, 4, 5, 6, 9, and DOT1L cellular assays**. PRMT6 assay: HEK293T cells were seeded in 12 well plates and transfected with 1 μg Flag-tagged PRMT6 WT or Mut(V86K/D88A) using jetPRIME® transfection reagent, following manufacturer instructions. After 4 h compounds were added and after 20 h cells were lysed in lysis buffer and analyzed in western blot for histone H4R3me2a levels normalized to H4[76].

PRMT4 assay: C2C12 cells were seeded in 12 well plates and next day treated with compounds for 48 h. After 48 h cells were lysed in lysis buffer and analyzed in western blot for BAF155R1064me2a levels normalized to BAF155[82].

PRMT5 assay: C2C12 cells were seeded in 12 well plates and next day treated with compounds for 48 h. After 48 h cells were lysed in lysis buffer and analyzed in western blot for SmBB'-Rme2s levels normalized to SmBB'[47].

PRMT1 assay: MCF7 cells were seeded in 12 well plates and next day treated with compounds for 48 h. After 48 h cells were lysed in lysis buffer and analyzed in western blot for histone H4R3me2a levels normalized to H4[76].

PRMT9 assay: HEK293T cells were seeded in 12 well plates and co-transfected with 0.9 μg FLAG-tagged PRMT9 and 0.1 μg GFP-tagged SAP145 WT or R508K mutant using jetPRIME® transfection reagent, following manufacturer instructions. After 4 h compounds were added and after 20 h cells were lysed in lysis buffer and analyzed in western blot for SAP145-Rme2s levels normalized to GFP[83].

DOT1L assay: THP-1 cells were seeded in 12 well plates and next day treated with compounds for 48 h. After 48 h cells were lysed in lysis buffer and analyzed in western blot for histone H3K79me2 levels normalized to H3[46].

**Western blot**. Total cell lysates or cellular fractions (as indicated) were resolved in 4–12% Bis-Tris Protein Gels (Invitrogen) and transferred in for 1.5 h (80 V) onto PVDF membrane (Millipore) in Tris-Glycine transfer buffer containing 20% MeOH and 0.05% SDS. Blots were blocked for 1 h in blocking buffer (5% milk in PBS) and incubated with primary antibodies in blocking buffer (5% BSA in PBST: 0.1% Tween-20 PBS) overnight at 4 °C. After five washes with PBST the blots were incubated with goat-anti-rabbit (IR800) and donkey anti-mouse (IR680) antibodies in Odyssey Blocking Buffer (LiCor) for 1 h at RT and washed five times with PBST. The signal was read on an Odyssey scanner (LiCor) at 800 and 700 nm. Band intensities for western blot analysis were determined using Image Studio Ver 5.2 (Licor). The uncropped blots are provided in the Source Data file.

**Knockdown**. Cells were transfected with 15 nM of either non-targeting siRNA or siRNA against PRMT7, PRMT4, or PRMT5 (Dharmacon) using Lipofectamine™ RNAiMAX, following manufacturer instructions. After 3 days, the protein levels were measured by western blot as described above.

**Cell growth and apoptosis assay after heat shock**. MEF WT and *Prmt7* KO cells were heat-shocked in a water bath for 20 min at 44 °C. For experiments with SGC3027 or SGC3027N, cells were pretreated with 3 μM compounds for 2 days before heat shock. Cell number was determined with Vybrant® DyeCycle™ Green, following manufacturer's instructions, 24 h after heat shock and apoptosis levels were determined with IncuCyte® Caspase-3/7 Reagent within 12 h after heat shock using IncuCyte™ ZOOM live cell imaging device (Essen Bioscience) and analyzed with IncuCyte™ ZOOM (2015A) software. Apoptosis levels and cell confluency were analyzed with IncuCyte™ ZOOM (2015A) software.

**Cell growth after bortezomib treatment**. MEF WT and *Prmt7* KO cells were treated with bortezomib for 4 or 24 h, and the confluency monitoring was started at 4 h after bortezomib (30 nM) treatment. For experiments with SGC3027 or SGC3027N, cells were pretreated with 4 μM compounds for 2 days before bortezomib treatment (30 nM). After 20 h bortezomib, SGC3027 or SGC3027N were removed and SGC3027 or SGC3027N were replaced. Cell confluency was monitored right after bortezomib addition using IncuCyte™ ZOOM live cell imaging device (Essen Bioscience) and analyzed with IncuCyte™ ZOOM (2015A) software.

**Cellular fractionation**. Cells were trypsinized and $1 \times 10^6$ cells were centrifuged at 400×*g* for 5 min at 4 °C. Cell pellets were resuspended in 200 μL of hypotonic buffer (10 mM HEPES pH 7.5, 10 mM KCl, 1.5 mM MgCl$_2$, 0.3 M Sucrose, 1 mM TCEP, 0.1% Triton X-100 and protease inhibitors). The cell suspensions were incubated on ice for 15 min followed by centrifugation at 1300×*g* for 5 min at 4 °C. The supernatants were collected and cleared by centrifugation at 18,000×*g* to produce the cytoplasmic fraction. The pellets were then washed in hypotonic buffer, centrifuged again, and resuspended in an equal volume of lysis buffer (as described above in PRMT7 cellular assay) to produce nuclear fraction. The fractions were analyzed by western blot as described above.

**Immunofluorescence**. C2C12 cells were electroporated (1650 V, 10 ms, 3 pulses) with 0.5 μg of PRMT7-FLAG plasmid using Neon transfection system (Life Technologies), following manufacturer instructions. Other cells were transfected with PRMT7-FLAG using X-tremeGene HP transfection reagent (Roche), following manufacturer instructions. The next day cells were washed with PBS, fixed with 4% PFA for 10 min, permeabilized with 0.1% Triton X-100/PBS for 5 min, blocked with 5% BSA in PBST (PBS, 0.1% Tween-20) for 1 h and incubated with anti-FLAG antibodies (1:1000) overnight at 4 °C. Next day cells were washed with PBST, incubated with anti-mouse Alexa Fluor 488 in blocking buffer (1:1000) and washed with PBST. Nuclei were labeled with Hoechst 33342 dye (ThermoFisher Scientific), following manufacturer instructions. The images were taken with EVOS FL Auto 2 Imaging System (ThermoFisher Scientific).

For the stress granule formation assessment, the immunofluorescence was performed in HeLa cells with cherry-G3BP1[84] as above using anti-TIAR and secondary anti-rabbit IgG Alexa647 antibodies. Cells were transfected with GFP-HSPA1 WT or R469K, and next day treated with 20 μM MG132 for 6 h. Stress granules were quantified on a percentage per cells basis counting the number of cells with at least one discrete G3BP1 foci that were also positive for TIAR from >100 cells. Three independent experiments were used for microscopy analysis performed with Quorum Spinning Disk Confocal microscope equipped with 405, 491, 561, and 642-nm lasers. Statistical significance was assessed using GraphPad Prism 7 software via Student's *t*-test (unpaired, 95% confidence interval) and one-way ANOVA with Tukey's post-hoc test. *p*-values < 0.05 were considered statistically significant.

**Immunoprecipitation**. HCT116 *PRMT7* KO (clone 94A) cells were co-transfected with GFP-tagged HSPA8/1 (WT or R469A mutant) and FLAG-tagged PRMT7

(WT or R44A mutant) at 1:10 ratio using JetPRIME transfection reagent (Poly-plus), following manufacturer's instructions. Cells were lysed in lysis buffer (20 mM Tris–HCl pH 8, 150 mM NaCl, 1 mM EDTA, 10 mM MgCl$_2$, 0.1% Triton X-100 and complete EDTA-free protease inhibitor cocktail (Roche)) for 20 min and centrifuged 18,000 × $g$ for 3 min. The supernatants were incubated with rabbit anti-GFP antibody (Invitrogen) overnight at 4 °C. Next day the antibody complexes were incubated with prewashed Dynabeads™ Protein G (ThermoFisher) for 2 h. Beads were washed in lysis buffer and proteins were eluted with 2 × SDS loading buffer and analyzed by western blot, as described above.

**LC-MS measurement of intracellular compounds concentration**. C2C12 cells were plated in 6-well plates (2 × 10$^6$ per well). Next day 3 μM of SGC3027 or SGC3027N was added to the cells and incubated for indicated times. After incu-bation, cells were washed with PBS, trypsinized, and cell pellets were collected by centrifugation at 500 × $g$ for 2 min. Pellets were mixed with 20 μL of acetonitrile, centrifuged for 1 min at 18,000 × $g$ and supernatants were collected and analyzed by LC–MS. To generate the standard curves, SGC8158 and SGC8158N compounds in two-fold dilution series from 0.025 to 25 μM were utilized. SGC3027 and SGC3027N compounds were also run to ensure the separation of the peaks and sufficient difference in the retention times. Standard curves were prepared in PBS. We spiked the PBS with 10 mM DMSO stock then did 2-fold dilution series for the calibration curve. The compounds were extracted using two volumes of acetonitrile (i.e. for each 20 μL of solution 40 μL acetonitrile was used). In the preliminary experiment during method development, we observed no ion suppression due to PBS; that means the chromatographic peak for same concentration was similar from water or PBS after acetonitrile extraction. However, we did not evaluate the extraction efficiency of these compounds from the cell matrix, hence the concentration reported were relative concentrations, not absolute. Chromatographic separations were carried out on an ACQUITY UPLC BEH C18 (2.1 × 50 mm, 1.7 μm) column. The mobile phase was 0.1% formic acid in water (solvent A) and 0.1% formic acid in acetonitrile (solvent B) at a flow rate of 0.4 mL min$^{-1}$. A gradient starting at 95% solvent A going to 5% in 4.5 min, holding for 0.5 min, going back to 95% in 0.5 min and equilibrating the column for 1 min was employed. A Waters SYNAPT G2-S MS equipped with an atmospheric pressure ionization source was used for MS analysis. In a typical MS acquisition setting, we used capillary voltage at 2.0 kV, sampling cone voltage at 20 V and the trap col-lision energy at 30.0 eV (detailed settings can be found in the Supplementary Table 7). MassLynx 4.1 software from Waters was used for data analysis with the QuanLynx module for quantification. Standard curves were generated by using the linear fit of mass peak areas and the known concentrations of SGC8158 and SGC8158N.

**HSPA8 ATPase assay and luciferase refolding assay**. ATPase assay was per-formed according to Cheng et al.[85] Briefly, purified HSPA8 or mutant HSPA8 (R469K) (10 μM) was incubated for 3 h at RT with or without purified full-length PRMT7 (0.5 μM) and with or without SAM (cold, 5 μM), as indicated, in buffer containing: 20 mM Tris–HCl (pH = 8.5), 5 mM DTT, 1 mM MgCl$_2$, 1 mM ATP, 0.01% Triton X-100. Before the ATPase assay, the samples were tested for mono-methylation levels in western blot. Processed HSPA8WT/MUT samples were diluted to 1 μM in assay buffer containing: 1.7 μM Hsp40 (Enzo), 0.017% Triton X-100, 100 mM Tris–HCl pH 7.4, 20 mM KCl and 6 mM MgCl$_2$. Fifteen μL of this mixture was added into each well of a 96-well plate and 10 μL of 2.5 mM ATP was added to start the reaction. The reaction was stopped after 0 and 1 h incubation at 37 °C with 50 μL of BIOMOL GREEN™ Reagent (Enzo Life Sciences). After 30 min incubation at RT the absorbance was measured at 620 nm. The phosphate con-centration was calculated from standard curve, prepared following manufacturer's instructions. As ATP had to be present for HSPA8 methylation reaction, the initial phosphate concentration (time 0) was measured for background subtraction.

Luciferase refolding assay[86] was performed in HEK293T cells transfected with HSPA1 WT or R469K. The next day cells were treated with 50 μg mL$^{-1}$ cyclohexamide and heated for 60 min at 45 °C to inactivate luciferase and, after 2 h recovery at 37 °C, luciferase activity was measured by using luciferase assay (Promega). Luciferase activity was normalized to unheated control samples.

**NanoBRET assay for HSPA8 co-chaperone association in cells**. HEK293T cells were plated in 96-well plates (2 × 10$^4$ per well) and 4 h later transfected with 0.01 μg C-terminally Nanoluc-tagged STIP1 (WT or K8A mutant) or STUB1 (WT or K30A mutant) and 0.09 μg of C-terminally Halo-tagged HSPA8 (WT or R469K mutant) or Halo tag alone using Xtreme gene HP transfection reagent (Roche), following manufacturer's instructions. Next day media was replaced with 80 μL of DMEM/F12 (no phenol red) +/− HaloTag® NanoBRET™ 618 Ligand (1 μL mL$^{-1}$, Promega) and 4 h later 20 μL of NanoBRET™ Nano-GloR Substrate (10 μL mL$^{-1}$ in DMEMF12 no phenol red, Promega) was added, and signal was read. Donor emission at 450 nm (filter: 450 nm/BP 80 nm) and acceptor emission at 618 nm (filter: 610 nm/LP) was measured within 10 min of substrate addition using CLARIOstar microplate reader (Mandel). Mean corrected NanoBRET ratios were determined by subtracting mean of 618/460 signal from cells without NanoBRET™ 618 Ligand × 1000 from mean of 618/460 signal from cells with NanoBRET™ 618 Ligand × 1000.

**CRISPR/Cas9 gRNA vector design for HCT116 cells**. For HCT116 cells, three guide RNAs were designed on PRMT7 locus. To generate gRNA expression vec-tors, the annealed oligonucleotide for each targeting site and annealed scaffold oligonucleotides were ligated into pENTER/U6 vector (Life Technologies). Cas9 expression vector was prepared previously[87]. Briefly, the Cas9 cDNA was syn-thesized by Eurofins genomics and inserted into the pCAGGS expression vector provided by Dr. J. Miyazaki (Osaka University, Osaka, Japan)[88]. Following guide RNA was used for generation of knockout cells: guide RNA/synthetic Oligonu-cleotide_sense/synthetic Oligonucleotide_antisense (21:GGGACTCTTGTCAAT GATGGCGG/caccGGGACTCTTGTCAATGATGGgtttta/ctctaaaacccatcattgaca agagtccc; 74:GGCATGGGTACTCCCACAGCGGG/caccGGCATGGGTACTC CCACAGCgtttta/ctctaaaacgctgtggggagtacccatgcc; 94:GGGCAGCTCTCCACGTC AACGGG/ caccGGGCAGCTCTCCACGTCAACGgtttta/ ctctaaaacgttgacgtggaga gctgcccc).

**CRISPR/Cas9-mediated genome editing**. HCT116 cells were seeded onto 6-well plates at a density of 40,000 cells per well, 24 h before transfection. Cells were transfected using Lipofectamine 2000 (Life Technologies) according to the man-ufacturer's instruction. A total of 3 μg Cas9 expression vector, 1 μg of gRNA expression vector, and 0.4 μg of pEBMultipuro (Wako Chemicals) as a transfection marker were transfected. After 48 h of transfection, 1 μg mL$^{-1}$ puromycin was added for selection. The colonies were isolated by limiting dilution. PRMT7 destruction for the isolated clones was confirmed by Sanger sequencing and wes-tern blotting.

**CRISPR for mouse C2C12 cells**. C2C12 PRMT7 KO clones were generated by cloning guide RNA (caccgGTCATGTAGCATGTCGGCAT/aaacATGCCGACA TGCTACATGAC) into LentiGuide-Puro vector (from Zhang lab, obtained from Addgene), following Zhang laboratory protocols. Lentivirus was produced using standard protocols. 48 h post transfection the supernatant was collected, filtered through 0.5 μm filter and used to infect C2C12 cells in presence of polybrene to final conc. of 8 μg mL$^{-1}$. After 24 h media was changed and after 2 days puromycin was added to final concentration of 2 μg mL$^{-1}$. 5 × 10$^4$ puromycin selected cells were electroporated with 0.5 μg Cas9-GFP plasmid using Neon transfection system (Life Technologies) (1650 V, 10 ms, 3 pulses). Next day, GFP-positive cells were sorted and plated in 96-well plates (1 cell per well). Cells were analyzed for PRMT7 expression and HSP70 monomethylation in western blot. The PRMT7 KO and PRMT7 catalytic mutant clones were genotyped by Sanger sequencing PCR-amplified TA-cloned genomic DNA. The following primers were used: forward CCA TCC AAT TGA GGT CAG CG and reverse TGG ACA TTC TTG AGC ACC TTA GT. The premature stop codon in exon 3 or 4 resulted in PRMT7 KO. The mutation in one of the alleles delY35, A35S in addition to premature codons in exon 3 or 4 of other alleles resulted in expression of catalytically inactive PRMT7 protein (Supplementary Table 5).

**Cell culture for methylome analysis**. Parental PRMT7 WT HCT116 cells were grown in DMEM (Wisent) and PRMT7 KO HCT116 cells (clone 74A) were grown in SILAC DMEM (w/o Arg and Lys, Wisent) supplemented with 146 mg L$^{-1}$ of heavy L-lysine hydrochloride (13C6,15N2, Sigma #608041) and 84 mg L$^{-1}$ of heavy L-arginine (13C6,15N4, Sigma-Aldrich #608033). Both media were supple-mented with 10% dialyzed FBS (Wisent), penicillin (100 U mL$^{-1}$) and strepto-mycin (100 μg mL$^{-1}$). Cells were labeled for 9 passages and incorporation of heavy amino acids was tested prior the LC–MS/MS experiment.

**Enrichment of monomethylarginine peptides**. The samples were enriched for monomethylated peptides with PTMScan® Mono-Methyl Arginine Motif [mme-RG] Kit (Cell Signaling) according to manufacturer's instructions. Briefly, cells were washed three times with PBS, lysed with 9 M urea buffer and centrifuged for 15 min at 20,000 × $g$ to remove cellular debris. The recovered proteins were quantified with a BCA Protein Assay Kit (ThermoFisher Scientific). For each replicate 5 mg of light and heavy amino acid-labeled cell lysate was combined, reduced with DTT (RT for 1 h), alkylated with iodoacetamide (RT for 15 min), digested with 100 μg of Lys-C (Wako, RT for 2 h), followed by 4-fold dilution with 20 mM HEPES pH = 8 and digested with 100 μg of Trypsin Gold (Promega) overnight at RT. The protein digests were acidified with trifluoroacetic acid to final concentration of 1%, purified using Sep-Pak C18 classic cartridge (Waters, #WAT051910) and lyophilized. Peptides were resuspended in IP buffer, cen-trifuged 5 min at 10,000 × $g$ at 4 °C and supernatant was incubated with anti-monomethyl arginine beads for 2 h at 4 °C. After 2 washes with 1× IP buffer, two washes with 10× diluted IP buffer and one wash with HPLC water, the peptides were eluted using 50 μl of 0.15% trifluoroacetic acid. The eluate was purified, concentrated using C18 spin tips (Pierce, #84850) and lyophilized. The peptides were digested again with 250 ng of Trypsin Gold (Promega) and purified using C18 spin tips.

**Mass spectrometric analysis of monomethylarginine peptides**. Mono-methylarginine peptides were analyzed by nanoLC-MS using a home-packed spray tip formed on a fused silica capillary column (0.75 μm internal diameter, 350 μm outer diameter) using a laser puller (Sutter Instrument Co., model P-2000). C18

reversed-phase material (Reprosil-Pur 120 C18-AQ, 3 μm, Dr. Maisch) in methanol was packed [15 (±1) cm] into the column using a pressure injection cell. An Eksigent 425 nano HPLC system (Sciex, Framingham, MA) was coupled to an Orbitrap Fusion Lumos (ThermoFisher Scientific, Waltham, MA). The LC gradient was delivered at 200 nL min$^{-1}$ and consisted of a ramp of 2–35% acetonitrile (0.1% formic acid) over 116 min, 35–80% acetonitrile (0.1% formic acid) over 19 min, 80% acetonitrile (0.1% formic acid) for 30 min, and then 7.5% acetonitrile for 29 min. The Orbitrap Fusion Lumos (Tune version 3.3) with Xcalibur (version 4.4) was operated in data-dependent acquisition (DDA) mode with survey scans performed at 120,000 resolution, AGC target of $5 \times 10^5$, with a maximum fill time of 50 ms, 400–1500 $m/z$ range. Top speed mode was used with a 1 s cycle time and 10 s dynamic exclusion. Fragment ions from MS/MS were detected in the orbitrap with 15,000 resolution, with an AGC target of $2 \times 10^5$, max fill time 35 ms. Charge states 2–6 were included with higher collisional dissociation (HCD) energy set at 32%. In addition to the 1 s data-dependent MS/MS, in every cycle a targeted list of precursors was collected with the same settings used in DDA, except the AGC target was $1 \times 10^5$ (targeted masses; 599.33, 604.33, 608.23, 606.33, 407.89, 403.22, 651.35, 657.29, 656.02, 652.62, and 611.34).

**Monomethyl arginine data analysis**. Raw files were searched using MaxQuant version 1.6.2.1 and the UP000005640 UniProt Release 2018_08 human database (Swiss-Prot reference containing 20,352 protein entries, downloaded on 24 October, 2018). PTM scores for monomethyl arginine were generated using the MaxQuant platform as previously described where site level occupancy was calculated by the ratio of modified peptide in two samples, the unmodified peptide version and the protein ratio[89]. Cysteine residues were searched as a fixed modification of +57.0215 Da, oxidized methionine residues as a variable modification of +15.9949 Da and deamidated asparagine residues as a variable modification of +0.9840 Da, and methylation of lysine or arginine residues as a variable modification of +14.0266 Da. Heavy SILAC labeling of lysine (K) and arginine (R) residues were set as variable modifications of +10 Da for heavy R and +8 Da for heavy K. All default settings for the 'Orbitrap' instrument type were used. This included mass tolerances of 20 ppm and 0.5 Da for MS1 and MS2 searches, respectively. Requantify was enabled and peptides were queried using trypsin/P cleavage constraints with a maximum of two missed cleavages sites. Match between runs was enabled. The peptide and protein false-discovery rate was set to 0.01 (1% FDR).

Peptide-level mean normalized $H/L$ ratios were first filtered for arginine monomethylated peptides occurring in at least two biological replicates, followed by significance testing using the limma package (v3.38.3) in R[90]. Significant hits were called as $H/L$ ratio of $<-1$ (knockout cells (H) relative to control (L)) and a Benjamini–Hochberg adjusted $p$-value of <0.01 ($n = 4$). Gene ontology enrichment analysis was performed using clusterProfiler (ver. 3.10.1)[91]. $p$-values from four independent replicates calculated by empirical Bayes moderated $t$-tests and adjusted using the Benjamini–Hochberg procedure as implemented in the Bioconductor package limma (v3.38.3)[90].

**PRMT7 chemical probe synthesis**. Experimental procedures and characterization data for chemical probe synthesis is described in Supplementary Methods section and illustrated in Supplementary Fig. 16.

**Reporting summary**. Further information on research design is available in the Nature Research Reporting Summary linked to this article.

## Data availability
All mass spectrometry raw files been deposited in the MassIVE repository housed at UCSD (https://massive.ucsd.edu/) with the accession number MSV000084773 [https://doi.org/doi:10.25345/C5GQ37]. Direct ftp download is available here ftp://massive.ucsd.edu/MSV000084773/. Raw input files are found within the updates folder. Known methylation sites were referenced from PhosphoSitePlus® v6.5.8 for Fig. 2c and Supplementary Table 4. The mPRMT7_SGC8158 structure has been deposited under the accession code PDB 6OGN. Data relating to Fig. 2c, Supplementary Fig. 2, and Supplementary Table 4 provided as Supplementary Data files. The source data underlying Figs. 1b, 2a, b, 3a–c, 4d–f, 5a, c, d, e, and 6a–e, h and Supplementary Figs. 1, 3, 5–15, Supplementary Tables 1–3 are provided as the Source Data file. All other data are available from the corresponding authors on reasonable request.

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

## Acknowledgements

The SGC is a registered charity (Number 1097737) that receives funds from AbbVie, Bayer Pharma AG, Boehringer Ingelheim, Canada Foundation for Innovation, Eshelman Institute for Innovation, Genome Canada through Ontario Genomics Institute [OGI-055], Innovative Medicines Initiative (EU/EFPIA) [ULTRA-DD Grant No. 115766], Janssen, Merck KGaA, Darmstadt, Germany, MSD, Novartis Pharma AG, Ontario Ministry of Research, Innovation and Science (MRIS), Pfizer, São Paulo Research Foundation-FAPESP, Takeda, and Wellcome [106169/ZZ14/Z]. D.D. receives support from a fellowship from the Natural Sciences and Engineering Research Council. M.T. and M.S.-O. are supported by Stand Up to Cancer Canada. A.L.C. and A.-C.G. are supported by the Canada Foundation for Innovation funding, by the Ontario Government, and by Genome Canada and Ontario Genomics (OGI-139). This work is based upon research conducted at the Northeastern Collaborative Access Team beamlines, which are funded by the National Institute of General Medical Sciences from the National Institutes of Health (P30 GM124165). The Eiger 16M detector on 24-ID-E beamline is funded by a NIH-ORIP HEI grant (S10OD021527). This research used resources of the Advanced Photon Source, a U.S. Department of Energy (DOE) Office of Science User Facility operated for the DOE Office of Science by Argonne National Laboratory under Contract No. DE-AC02-06CH11357. We thank Kumar Singh Saikatendu, Charles E. Grimshaw, Nobuo Cho, Toshiyuki Nomura, and Atsushi Kiba for suggestions and support throughout the project as well as Michiko Tawada and Sachiko Itono for suggestions and chemistry support, and Yoshihiko Hirozane, Masato Yoshikawa, and Satoshi Sogabe for the kinase selectivity data. We thank Alma Seitova, Ashley Hutchinson, and Hong Zheng for recombinant protein production, and Aiping Dong for crystallography support.

## Author contributions

M.M.S. performed compound cellular activity determination, PRMT7 knockdowns, CRISPR knockouts, cellular fractionation, cellular selectivity testing, immunoprecipitation, bortezomib heat shock, sample preparation for proteomic analysis, and functional experiments. Y.I. designed the biological studies at Takeda. S.O. performed early compound cellular activity determination, designed and validated PRMT7 cell assay, performed PRMT7 knockdowns, heat shock experiments, immunoprecipitation verification of HSP70 substrates, and knockdown HSP70. N.S. led medicinal chemistry activities, and contributed to design and syntheses of a series of SAM-mimetic derivatives including SGC8158 and SGC8172. F.L., M.E., and C.S. performed in vitro compound characterization. S.A., A.L.C., M.S.-O., E.B., and K.H. performed and analyzed proteomics experiments for substrate identification. D.D. performed CRISPR knockout C2C12 clone characterization in, and contributed to proteomics analysis and figure generation. L.H. contributed the co-crystal structures and H.F. contributed design and syntheses of a series of SAM-mimetic derivatives including SGC8158 and SGC8172. R.H. performed structural analysis of HSP70 proteins. T.S. established PRMT7 KO HCT116 cells. D.M., C.Z., and R.A.-A. synthesized SGC3027 and SGC3027N. A.A. performed analysis of prodrug conversion in cells. S.T. performed the immunoblot analysis. S.R. provided *Prmt7* KO MEF and advice on PRMT7 biology, unpublished data. M.T. oversaw the project and collaboration with SGC, made research strategy/direction, and suggestions for biological evaluations of the compounds. M.T. and A.-C.G. oversaw proteomics experiments. C.H.A. oversaw the project and collaboration with Takeda. M.V. oversaw the in vitro characterization of compounds and substrates. P.J.B. oversaw the collaboration and prodrug-based chemistry strategy. H.N. oversaw the medicinal chemistry program and contributed to the initial hit finding process for identification of SGC0911. D.B.L. oversaw the cellular characterization of compounds and functional experiments. M.M.S., S.A., D.D., L.H., Y.I., N.S., R.H., H.N., M.V., C.H.A., P.J.B., and D.B.L. wrote the manuscript.

## Competing interests

Y.I., N.S., H.F., T.S., K.H., S.T., M.T., and H.N. are current or former employees of Takeda. M.M.S., S.O., F.L., L.H., S.A., M.E., D.D., R.H., C.C.S., C.H.A., M.V., P.J.B., and D.B.-L. are current or former employees of the University of Toronto. The Structural Genomics Consortium is funded in part by Takeda. All other authors declare no competing interests.
