## [Peer Review File · Nature Communications]

Reviewers' comments:

Reviewer #1 (Remarks to the Author):

In this revised manuscript, Szewczyk et al reported and validated their studies on probe development and new cellular function for PRMT7. This is elegant work. Several novel findings were reported: proteomic profiling of PRMT7 substrates; role of PRMT7 in HSP70 regulation and in stress response, and identification of SGC3027 as a cell active selective inhibitor of PRMT7. Methylation of HSP70 by PRMT7 is a neat example showing substrate methylation regulation at the substrate level. Reviewers' concerns were mostly addressed. I would thus recommend acceptance of this manuscript for publication.

Fig. 5a, is the methylation assay done in the presence of ATP?

In the future, it would be interesting to determine the structural basis of how R469 methylation affects HSP70 structure and activity.

Reviewer #4 (Remarks to the Author):

As a late reviewer of this manuscript, I was tasked with providing an assessment of author responses to the comments raised by reviewer #3. After carefully reading the reviewer's comments and the authors's detailed responses and manuscript modifications, it is my opinion that the authors have carefully and extensively addressed each of reviewer #3's concerns. They added significant new data that support their original hypotheses and they provide clear answers as to why it is beyond the scope of the current work to fully address a couple of the suggestions offered by the reviewer. I think they have made a strong case for PRMT7 arginine methylation of Hsp70 in the ligand binding cleft and the importance of this methylation for Hsp70 holdase/refolding activity. Methylation occurs only in the ATP-bound ("open") state of Hsp70 (although ATPase activity of the chaperone is

not affected), and this is entirely consistent with ATP-bound Hsp70 recognizing unfolded motifs in client proteins. Identification of a cell-permeable inhibitor of PRMT7 (as well as a negative control compound) will allow for careful study of the mechanistic basis underlying the impact of this methylation on substrate binding and ultimately could be of clinical significance in disrupting proteostasis in cancer cells.

Reviewer #5 (Remarks to the Author):

In this manuscript, the authors reported a chemical probe SGC8158 (prodrug SGC3027) as a potent and selective inhibitor of PRMT7. Subsequently, they employed SILAC-based quantitative proteomic approach to screen PRMT7 substrates and identified HSP70 proteins as candidates. The methylation site R469 as PRMT7 substrates were further validated by a series of molecular biology experiments at cellular level and in vitro. They then found that the open form of HSP70 was required for PRMT7 methylation, and the methylation of R469 on HSP70 can be inhibited by SGC8158, which is involved in modulation of cellular proteostasis and stress response. As requested by the Editor, I exclusively reviewed the contents including proteomic analysis of arginine monomethylation and mass spectrometric analysis of small molecules. Overall, the technical quality is good, but it can still be improved after addressing the questions below.

1. Genetic manipulation of cells might cause protein alterations. For proteomic quantitation of PTMs, the false-positive quantitation results could be obtained due to alteration of protein levels, instead of change of PTMs. Therefore, additional experiment on quantitation of proteins, without enrichment of the methylated tryptic peptides, is better to be performed to filter out the false quantitation caused by protein alteration after PRMT7 knockout.
2. (Page 35, Line 970) More details about statistical analysis should be added in the figure legend. Student t test? Are the p values adjusted? By what multiple testing method? How many replicates?
3. (Page 7, Line 152) There is no statement on performing PRM. The PRM experiment is only mentioned in one sentence at Page 27, Line 716. Is it to validate the quantitation of the peptides listed in Supplementary figure 5? Also, how many proteins are quantifiable? It will be helpful to attach an Excel file showing the detailed quantitative information of all the proteins.
4. (Page 7, Line 149) typo: "24 unique proteins" should be "24 unique peptides".
5. (Page 27, Line 712) The mass spectrometric settings in Supplementary Table 7 and 8 can be summarized in the Methods section rather than be listed as tables. Please refer to Coon et al's publication (Methods in Page 10 of Coon et al. Nat Commun, 2017, 8, 15571)

6. (Page 27, Line 721) More details need to be included for the data analysis. What mass error of MS1 and MS2? How the quantitative information of peptide level be converted to site level? Please refer to Coon's publication as well.

7. (Page 24, Line 615) Details of mass spectrometric settings are needed.

Reviewer #6 (Remarks to the Author):

Overall, the structural analysis described in the manuscript is of high quality. The crystallographic statistics are good. The crystals of the complex with SGC8158 were obtained by soaking SAH-bound PRMT7 crystals in SGC8158. The omit map shown in Suppl. Figure 4 does not cover the whole inhibitor, however the ribosyl moiety is very convincing. The biphenylmethylamine moiety is less clear. One of the reviewers was concerned by the possibility that the density assigned to the biphenylmethylamine moiety could in fact correspond to the unmodelled side chain of W314. From the provided figures it is not obvious to judge. Perhaps, the authors could provide the Suppl Figure 4 in the same orientation as in Fig1 d,e,f and label the shown residues. In addition providing a panel of the SAH-PRMT7 complex showing the same omit map calculated without SAH and His313 and Trp314 residues could help clarifying this issue.

Dear Reviewers,

We would like to thank you for your time and helpful comments as well as suggestions. We considered them very carefully, added the requested information and made changes to the manuscript that will improve the scientific rigor of experimental conclusions, facilitate communication of the results, and provide the evidence needed for the SGC3027 chemical probe compound utility. Below is the detailed list addressing the suggestions and concerns. The changes in the manuscript text are indicated in red. We hope that the revisions will meet the expectations of the reviewers and look forward to your response.

Kind regards,
Dalia

Reviewer #1 (Remarks to the Author):

In this revised manuscript, Szewczyk et al reported and validated their studies on probe development and new cellular function for PRMT7. This is elegant work. Several novel findings were reported: proteomic profiling of PRMT7 substrates; role of PRMT7 in HSP70 regulation and in stress response, and identification of SGC3027 as a cell active selective inhibitor of PRMT7. Methylation of HSP70 by PRMT7 is a neat example showing substrate methylation regulation at the substrate level. Reviewers' concerns were mostly addressed. I would thus recommend acceptance of this manuscript for publication.

Fig. 5a, is the methylation assay done in the presence of ATP?

In the future, it would be interesting to determine the structural basis of how R469 methylation affects HSP70 structure and activity.

We thank the reviewer for the kind comments and suggestions. The in vitro methylation in Fig. 5a was performed in the presence of ATP that is now clarified in the figure legend. We do hope that our work and the tool compound will benefit the scientific community in understanding the structural implications of R469 methylation of HSP70. While presently we do not have the mandate to work on this ourselves we would support anyone who would be interested in this exciting opportunity. We also believe that our data contribute to a better understanding of protein dynamics that is an important aspect of posttranslational modifications and protein

function.

Reviewer #4 (Remarks to the Author):

As a late reviewer of this manuscript, I was tasked with providing an assessment of author responses to the comments raised by reviewer #3. After carefully reading the reviewer's comments and the authors's detailed responses and manuscript modifications, it is my opinion that the authors have carefully and extensively addressed each of reviewer #3's concerns. They added significant new data that support their original hypotheses and they provide clear answers as to why it is beyond the scope of the current work to fully address a couple of the suggestions offered by the reviewer. I think they have made a strong case for PRMT7 arginine methylation of Hsp70 in the ligand binding cleft and the importance of this methylation for Hsp70 holdase/refolding activity. Methylation occurs only in the ATP-bound ("open") state of Hsp70 (although ATPase activity of the chaperone is not affected), and this is entirely consistent with ATP-bound Hsp70 recognizing unfolded motifs in client proteins. Identification of a cell-permeable inhibitor of PRMT7 (as well as a negative control compound) will allow for careful study of the mechanistic basis underlying the impact of this methylation on substrate binding and ultimately could be of clinical significance in disrupting proteostasis in cancer cells.

We thank the reviewer for the time and expertise in evaluating our work as well as for insightful comments.

Reviewer #5 (Remarks to the Author):

In this manuscript, the authors reported a chemical probe SGC8158 (prodrug SGC3027) as a potent and selective inhibitor of PRMT7. Subsequently, they employed SILAC-based quantitative proteomic approach to screen PRMT7 substrates and identified HSP70 proteins as candidates. The methylation site R469 as PRMT7 substrates were further validated by a series of molecular biology experiments at cellular level and in vitro. They then found that the open form of HSP70 was required for PRMT7 methylation, and the methylation of R469 on HSP70 can be inhibited by SGC8158, which is involved in modulation of cellular proteostasis and stress response. As requested by the Editor, I exclusively reviewed the contents including proteomic analysis of arginine monomethylation and mass spectrometric analysis of small molecules. Overall, the technical quality is good, but it can still be improved after addressing the questions below.

1. Genetic manipulation of cells might cause protein alterations. For proteomic quantitation

of PTMs, the false-positive quantitation results could be obtained due to alteration of protein levels, instead of change of PTMs. Therefore, additional experiment on quantitation of proteins, without enrichment of the methylated tryptic peptides, is better to be performed to filter out the false quantitation caused by protein alteration after PRMT7 knockout.

We thank the reviewer for this important point. We have added a complete analysis of the input proteins (Supplementary data 3 to be included with the manuscript) and updated the Supplementary Table 4 containing the abbreviated list of methylated proteins with the information of the matching input protein levels. These levels were obtained from the bottom-up proteomics on the input samples (before immunoprecipitation). The data analysis indicates that the differentially methylated input protein levels do not change significantly using the same significance cut-off criteria as for the arginine monomethylation analysis (significance cut-offs of H/L ratio < -1 and adjusted p-value < 0.01 (n=4)). The text was updated with the following sentence on page 7:

The analysis of total protein levels in PRMT7 KO and WT cells indicated no significant change in protein abundance for the differentially methylated peptides (Supplementary Table 4).

Although the total levels of PRMT7 methylated proteins were not significantly altered, we did detect other proteins that were affected by PRMT7 knockout, please see below. As the manuscript is focused on PRMT7 inhibition and its direct targets, we did not include this data analysis, however, the full dataset is available. The Supplementary Table 4 heading was expanded to direct the reader to the full set of methylated proteins (Supplementary data 2) and total proteome analysis (Supplementary data 3).

Input proteome analysis: Differential expression analysis of input samples, using same threshold as IPs - $\log_2 H/L < -2$ & adjusted P-value < 0.01, identifies 71 down-regulated proteins in KO condition. None of these are found in the differentially methylated set.

2. (Page 35, Line 970) More details about statistical analysis should be added in the figure legend. Student t test? Are the p values adjusted? By what multiple testing method? How many replicates?

We thank the reviewer for bringing up this important point that we based our conclusions on. We have clarified and expanded the analysis description in Fig 2 legend (page 36) and methods (page 28) with the following text.

Volcano plot showing Log₂ Heavy/Light ratio of SILAC-labelled monomethyl arginine peptides from WT (L, unlabelled) relative to PRMT7 KO (H, heavy RK labelled) HCT116 cells. Dashed lines represent significance cut-offs of H/L ratio < -1 and adjusted p-value < 0.01 (n=4). Labelled points, further highlighted in red, correspond to reported Rme1 sites found in the PhosphoSitePlus database³⁰. P-values from four independent replicates calculated by empirical Bayes moderated t-tests and adjusted using the Benjamini-Hochberg procedure as implemented in the Bioconductor package limma (v3.38.3)⁸⁹

3. (Page 7, Line 152) There is no statement on performing PRM. The PRM experiment is only mentioned in one sentence at Page 27, Line 716. Is it to validate the quantitation of the peptides listed in Supplementary figure 5? Also, how many proteins are quantifiable? It will be helpful to attach an Excel file showing the detailed quantitative information of all the proteins.

We thank the reviewer for requesting clarification. The DDA was used in combination with PRM (targeted masses) so that in every run a list of HSPA8 (our main interest) peptides was targeted to ensure MS2 quantitation. The methods text on page 27 was updated. We also include an excel file (Supplementary data 3) with the protein quantitative information as per response to comment 1. There were 2556 proteins identified in the input (before immunoprecipitation) of which 2131 quantifiable. There were 346 peptides Rme belonging to 193 proteins identified in the immunoprecipitated samples. These lists are provided as supplementary data files 2 and 3 in addition to the Supplementary Table 4. The text on page 7 was also clarified (please see the text below for point 4).

4. (Page 7, Line 149) typo: "24 unique proteins" should be "24 unique peptides".

We thank the reviewer for flagging this confusing point. In Supplementary table 4, significantly differentially methylated peptides are shown by the protein accession number and gene name and some of the peptides, for example, EIF4G1 (3 peptides) are present in the same protein hit, same accession number. The text (page 7) was reworded for better clarity.

Wild-type (WT) and PRMT7 knockout (KO) HCT116 cells were subjected to SILAC (Stable Isotope Labeling by/with Amino acids in Cell culture) and monomethyl arginine immunoprecipitation followed by mass spectrometry analysis that included a targeted list of HSPA8 peptides (to

ensure MS2 quantitation) within the DDA cycle. Twenty-nine significantly differentially methylated peptides representing 24 unique proteins were identified. Twenty-one peptides (from 18 proteins) were previously reported as arginine methylated³⁰ (highlighted in **Fig. 2c, Supplementary Table 4**). The analysis of total protein levels in PRMT7 KO and WT cells indicated no significant change in protein abundance for the differentially methylated peptides (**Supplementary Table 4**).

5. (Page 27, Line 712) The mass spectrometric settings in Supplementary Table 7 and 8 can be summarized in the Methods section rather than be listed as tables. Please refer to Coon et al's publication (Methods in Page 10 of Coon et al. Nat Commun, 2017, 8, 15571)

We thank the reviewer for this suggestion. The tables have been converted to text as requested, please see page 27-28:

Mass Spectrometric analysis of mono-methylarginine peptides

Mono-methylarginine peptides were analyzed by nano-LCMS using a home-packed spray tip formed on a fused silica capillary column (0.75 μm internal diameter, 350 μm outer diameter) using a laser puller (Sutter Instrument Co., model P-2000). C18 reversed-phase material (Reprosil-Pur 120 C18-AQ, 3 μm , Dr. Maisch) in methanol was packed [15 (\pm 1) cm] into the column using a pressure injection cell. An Eksigent 425 nano HPLC system (Sciex, Framingham, MA) was coupled to an Orbitrap Fusion Lumos (Thermo Fisher Scientific, Waltham, MA). The LC gradient was delivered at 200 nl/m and consisted of a ramp of 2–35% acetonitrile (0.1% formic acid) over 116 m, 35–80% acetonitrile (0.1% formic acid) over 19 m, 80% acetonitrile (0.1% formic acid) for 30 m, and then 7.5% acetonitrile for 29 m. The Orbitrap Fusion Lumos (Tune version 3.3) with Xcalibur (version 4.4) was operated in data-dependent acquisition (DDA) mode with survey scans performed at 120,000 resolution, AGC target of 5×10^5 , with a maximum fill time of 50ms, 400-1500m/z range. Top speed mode was used with a 1s cycle time and 10s dynamic exclusion. Fragment ions from MS/MS were detected in the orbitrap with 15,000 resolution, with an AGC target of 2×10^5 , max fill time 35ms. Charge states 2-6 were included with higher collisional dissociation (HCD) energy set at 32%. In addition to the 1s data dependent MS/MS, in every cycle a targeted list of precursors was collected with the same settings used in DDA, except the AGC target was 1×10^5 (targeted masses; 599.33, 604.33, 608.23, 606.33, 407.89, 403.22, 651.35, 657.29, 656.02, 652.62, and 611.34).

6. (Page 27, Line 721) More details need to be included for the data analysis. What mass error of MS1 and MS2? How the quantitative information of peptide level be converted to site level? Please refer to Coon's publication as well.

We thank the reviewer for this point. MS1 and MS2 mass errors were left at 20ppm and 0.5 Da, respectively, the default settings in MaxQuant and now they are clearly stated in the methods text on page 28 that have been expanded to include more details:

Mass Spectrometric Data Analysis

Raw files were searched and quantified using Maxquant version 1.6.2.1 using the UP000005640 Uniprot human database (Swiss-Prot reference containing 20,352 protein entries, downloaded on 24 October, 2018). PTM scores for Rme1 were generated using the MaxQuant platform as previously described and site level occupancy is calculated by the ratio of modified peptide in two samples, the unmodified peptide version and the protein ratio⁸⁹. Cysteine residues were searched as a fixed modification of +57.0215 Da, oxidized methionine residues as a variable modification of +15.9949 Da and deamidated asparagine residues as a variable modification of +0.9840, and methylation of lysine or arginine residues as a variable modification of +14.0266 Da. Heavy SILAC labeling of lysine (K) and 724 arginine (R) residues were set as variable modifications of +10Da for heavy R and +8Da for heavy K. Mass tolerances were set to 20ppm and 0.5 Da for MS1 and MS2 searches, respectively. Re-quantify was enabled and peptides were queried using trypsin/P cleavage constraints with a maximum of two missed cleavages sites. Match between runs was enabled. The peptide and protein false-discovery rate was set to 0.01. All other parameters were default settings.

Peptide-level mean normalized H/L ratios were first filtered for arginine monomethylated peptides occurring in at least two biological replicates, followed by significance testing using the limma package (v3.38.3) in R⁸⁹. Significant hits were called as H/L ratio of less than -1 (knockout cells (H) relative to control (L)) and a Benjamini-Hochberg adjusted p-value of less than 0.01 (n=4). Gene ontology enrichment analysis was performed using clusterProfiler (ver. 3.10.1)⁹⁰. *P-values from four independent replicates calculated by empirical Bayes moderated t-tests and adjusted using the Benjamini-Hochberg procedure as implemented in the Bioconductor package limma (v3.38.3)*⁸⁹

7. (Page 24, Line 615) Details of mass spectrometric settings are needed.

We thank the reviewer for bringing this up. The above-mentioned methods relate to the intracellular measurements of the compounds. We have added the following information (highlighted in red) that indicated how the compound concentrations were determined and the methods as well as the software used.

C2C12 cells were plated in 6 well plates (2 x 10⁶/well). Next day 3 μM of SGC3027 or SGC3027N was added to the cells and incubated for indicated times. After incubation, cells were washed with PBS, trypsinized and cell pellets were collected by centrifugation at 500 x g for 2 min. Pellets were mixed with 20 μl of acetonitrile, centrifuged for 1 min at 18,000 x g and supernatants were collected and analyzed by LCMS. *To generate the standard curves SGC8158 and SGC8158N compounds in two-fold dilution series from 0.025 to 25 μM were utilized. SGC3027 and SGC3027N compounds were also run to ensure the separation of the peaks and sufficient difference in the retention time.* Chromatographic separations were carried out on an ACQUITY UPLC BEH C18 (2.1 X 50 mm, 1.7 μm) column. The mobile phase was 0.1% formic acid in water (solvent A) and 0.1% formic acid in acetonitrile (solvent B) at a flow rate of 0.4 mL/min. A gradient starting at 95% solvent A going to 5% in 4.5 min, holding for 0.5 min, going back to 95% in 0.5 min and equilibrating the column for 1 min was employed. A Waters Xevo QToF or a Waters

SYNAPT G2-S MS equipped with an atmospheric pressure ionization source was used for MS analysis. MassLynx 4.1 software from Waters was used for data analysis with the QuanLynx module for quantification. Standard curves were generated by using the linear fit of mass peak areas and the known concentrations of SGC8158 and SGC8158N.

Reviewer #6 (Remarks to the Author):

Overall, the structural analysis described in the manuscript is of high quality. The crystallographic statistics are good. The crystals of the complex with SGC8158 were obtained by soaking SAH-bound PRMT7 crystals in SGC8158. The omit map shown in Suppl. Figure 4 does not cover the whole inhibitor, however the ribosyl moiety is very convincing. The biphenylmethylamine moiety is less clear. One of the reviewers was concerned by the possibility that the density assigned to the biphenylmethylamine moiety could in fact correspond to the unmodelled side chain of W314. From the provided figures it is not obvious to judge. Perhaps, the authors could provide the Suppl Figure 4 in the same orientation as in Fig1 d,e,f and label the shown residues. In addition providing a panel of the SAH-PRMT7 complex showing the same omit map calculated without SAH and His313 and Trp314 residues could help clarifying this issue.

We thank the reviewer for helping us bring clarity to the placement of the compound in PRMT7. As requested, we provide Suppl Figure 4 panels with the same orientation as Fig 1d and the residue labels. We also include SAH-PRMT7 complex with the omit map calculated without SAH and H313, W314, indicated in panel d, please also compare panels b and c.